# Sparse dimensionality reduction approaches in Mendelian randomisation with highly correlated exposures

**Vasileios Karageorgiou[1,2]\*, Dipender Gill[1,3,4], Jack Bowden[2,4†], Verena Zuber[1†]**

[1]Department of Epidemiology and Biostatistics, School of Public Health, Faculty of Medicine, Imperial College London, London, United Kingdom; [2]University of Exeter, Exeter, United Kingdom; [3]Department of Clinical Pharmaceutics and Therapeutics, Institute for Infection and Immunity, St George's, University of London, London, United Kingdom; [4]Genetics Department, Novo Nordisk Research Centre Oxford, Oxford, United Kingdom

**Abstract** Multivariable Mendelian randomisation (MVMR) is an instrumental variable technique that generalises the MR framework for multiple exposures. Framed as a regression problem, it is subject to the pitfall of multicollinearity. The bias and efficiency of MVMR estimates thus depends heavily on the correlation of exposures. Dimensionality reduction techniques such as principal component analysis (PCA) provide transformations of all the included variables that are effectively uncorrelated. We propose the use of sparse PCA (sPCA) algorithms that create principal components of subsets of the exposures with the aim of providing more interpretable and reliable MR estimates. The approach consists of three steps. We first apply a sparse dimension reduction method and transform the variant-exposure summary statistics to principal components. We then choose a subset of the principal components based on data-driven cutoffs, and estimate their strength as instruments with an adjusted $F$-statistic. Finally, we perform MR with these transformed exposures. This pipeline is demonstrated in a simulation study of highly correlated exposures and an applied example using summary data from a genome-wide association study of 97 highly correlated lipid metabolites. As a positive control, we tested the causal associations of the transformed exposures on coronary heart disease (CHD). Compared to the conventional inverse-variance weighted MVMR method and a weak instrument robust MVMR method (MR GRAPPLE), sparse component analysis achieved a superior balance of sparsity and biologically insightful grouping of the lipid traits.

**\*For correspondence:**
vk282@exeter.ac.uk

†These authors contributed equally to this work

## Editor's evaluation

This paper investigated the identification of causal risk factors on health outcomes. It applies sparse dimension reduction methods on highly correlated traits in the Mendelian randomization framework. The implementation of this method helps to identify risk factors when given high dimensional traits data.

## Introduction

Mendelian randomisation (MR) is an epidemiological study design that uses genetic variants as instrumental variables (IVs) to investigate the causal effect of a genetically predicted exposure on an outcome of interest (*Smith and Ebrahim, 2003*). In a randomised controlled trial (RCT) the act of randomly allocating patients to different treatment groups precludes the existence of systematic confounding between the treatment and outcome and therefore provides a strong basis for causal

inference. Likewise, the alleles that determine a small proportion of variation of the exposure in MR are inherited randomly. We can therefore view the various genetically proxied levels of a lifelong modifiable exposure as a 'natural' RCT, avoiding the confounding that hinder traditional observational associations. Genetically predicted levels of an exposure are also less likely to be affected by reverse causation, as genetic variants are allocated before the onset of the outcomes of interest.

When evidence suggests that multiple correlated phenotypes may contribute to a health outcome, multivariable MR (MVMR), an extension of the basic univariable approach can disentangle more complex causal mechanisms and shed light on mediating pathways. Following the analogy with RCTs, the MVMR design is equivalent to a factorial trial, in which patients are simultaneously randomised to different combinations of treatments (*Burgess and Thompson, 2015*). An example of this would be investigation into the effect of various lipid traits on coronary heart disease (CHD) risk (*Burgess and Harshfield, 2016*). While MVMR can model correlated exposures, it performs suboptimally when there are many highly correlated exposures due to multicollinearity in their genetically proxied values. This can be equivalently understood as a problem of conditionally weak instruments (*Sanderson et al., 2019*) that is only avoided if the genetic instruments are strongly associated with each exposure conditionally on all the other included exposures. An assessment of the extent to which this assumption is satisfied can be made using the conditional *F*-statistic, with a value of 10 for all exposures being considered sufficiently strong (*Sanderson et al., 2019*). In settings when multiple highly correlated exposures are analysed, a set of genetic instruments are much more likely to be conditionally weak instruments. In this event, causal estimates can be subject to extreme bias and are therefore unreliable. Estimation bias can be addressed to a degree by fitting weak instrument robust MVMR methods (*Sanderson et al., 2020*; *Wang et al., 2021*), but at the cost of a further reduction in precision. Furthermore, MVMR models investigate causal effects for each individual exposure, under the assumption that it is possible to intervene and change each one whilst holding the others fixed. In the high-dimensional, highly correlated exposure setting, this is potentially an unachievable intervention in practice.

Our aim in this paper is instead to use dimensionality reduction approaches to concisely summarise a set of highly correlated genetically predicted exposures into a smaller set of independent principal components (PCs). We then perform MR directly on the PCs, thereby estimating their effect on health outcomes of interest. We additionally suggest employing sparsity methods to reduce the number of exposures that contribute to each PC, in order to improve their interpretability in the resulting factors.

Using summary genetic data for multiple highly correlated lipid fractions and CHD (*Kettunen et al., 2016*; *Nelson et al., 2017*), we first illustrate the pitfalls encountered by the standard MVMR approach. We then apply a range of sparse principal component analysis (sPCA) methods within an MVMR framework to the data. Finally, we examine the comparative performance of the sPCA approaches in a detailed simulation study, in a bid to understand which ones perform best in this setting.

## Results

### Workflow overview

Our proposed analysis strategy is presented in *Figure 1*. Using summary statistics for the single-nucleotide polymorphism (SNP)-exposure ($\hat{\gamma}$) and SNP-outcome ($\hat{\Gamma}$) association estimates, where $\hat{\gamma}$ (dimensionality 148 SNPs × 97 exposures) exhibits strong correlation, we initially perform a PCA on $\hat{\gamma}$. Additionally, we perform multiple sPCA modalities that aim to provide sparse loadings that are more interpretable (block 3, *Figure 1*). The choice of the number of PCs is guided by permutation testing or an eigenvalue threshold. Finally, the PCs are used in place of $\hat{\gamma}$ in an IVW MVMR meta-analysis to obtain an estimate of the causal effect of the PC on the outcome. Similar to PC regression and in line with unsupervised methods, the outcome (SNP-outcome associations ($\hat{\Gamma}$) and corresponding standard error ($SE_{\hat{\Gamma}}$)) is not transformed by PCA and is used in the second-step MVMR in the original scale. In the real data application and in the simulation study, the best balance of sparsity and statistical power was observed for the method of sparse component analysis (SCA) (*Chen and Rohe, 2021*). This favoured method and the related steps are coded in an *R* function and are available at GitHub (https://github.com/vaskarageorg/SCA_MR/, copy archived at *Karageorgiou, 2023*).

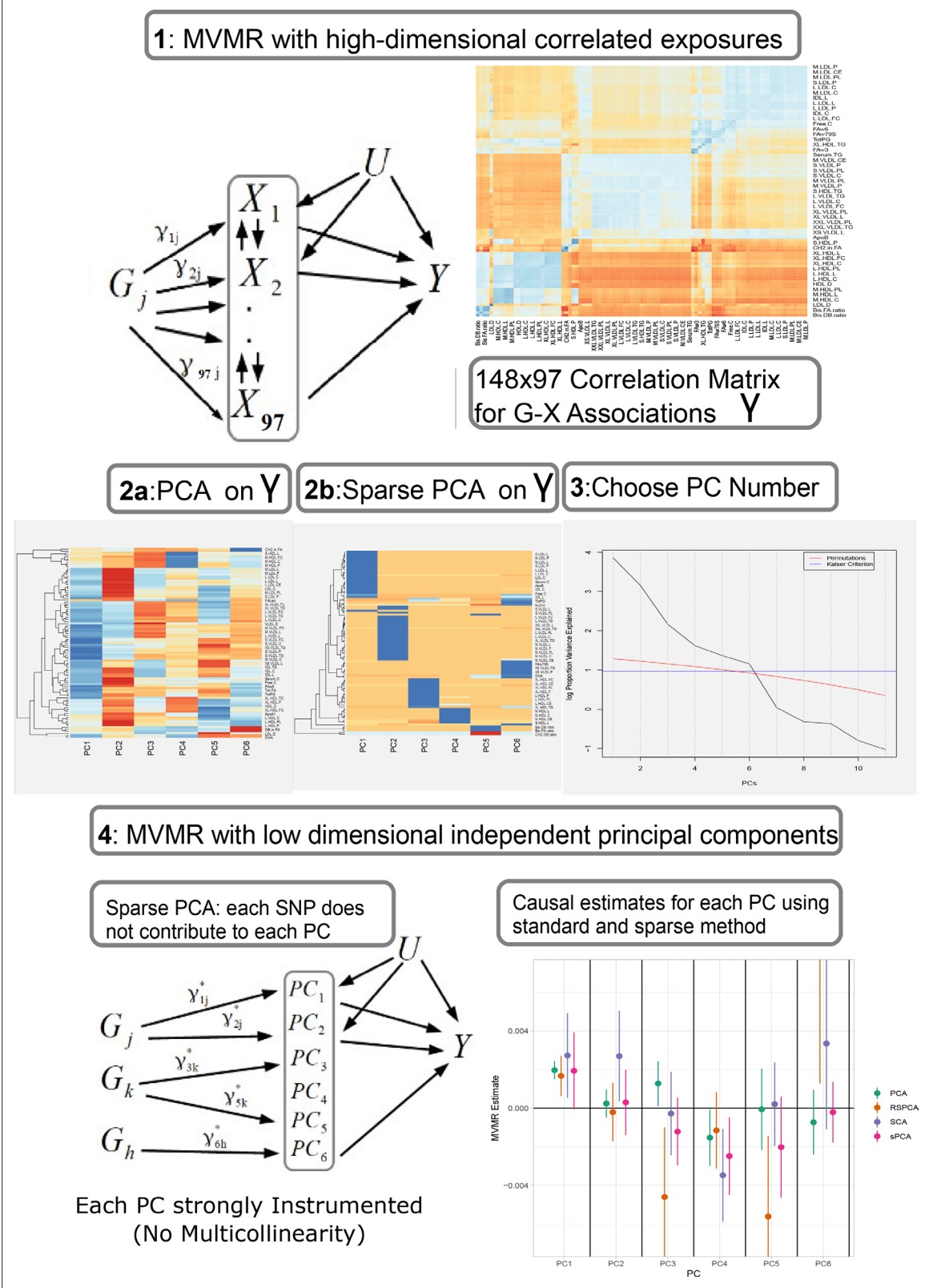

**Figure 1.** Proposed workflow. Step 1: MVMR on a set of highly correlated exposures. Each genetic variant contributes to each exposure. The high correlation is visualised in the similarity of the single-nucleotide polymorphism (SNP)-exposure associations in the correlation heatmap (top right). Steps 2 and 3: PCA and sparse PCA on $\hat{\gamma}$. Step 4. MVMR analysis on a low dimensional set of principal components (PCs). X: exposures; Y: outcome; k: number of exposures; PCA: principal component analysis; MVMR: multivariable Mendelian randomisation.

**Table 1.** Univariable Mendelian randomisation (MR) results for the Kettunen dataset with coronary heart disease (CHD) as the outcome.
Positive: positive causal effect on CHD risk; Negative: negative causal effect on CHD risk.

|  | Positive | Negative |
|---|---|---|
| VLDL | AM.VLDL.C, M.VLDL.CE, M.VLDL.FC, M.VLDL.L,M.VLDL.P, M.VLDL.PL, M.VLDL.TG, XL.VLDL.L,XL.VLDL.PL, XL.VLDL.TG, XS.VLDL.L, XS.VLDL.P, XS.VLDL.PL,XS.VLDL.TG, XXL.VLDL.L, XXL.VLDL.PL,L.VLDL.C, L.VLDL.CE, L.VLDL.FC, L.VLDL.L, L.VLDL.P,L.VLDL.PL, L.VLDL.TG, SVLDL.C, S.VLDL.FC,S.VLDL.L, S.VLDL.P, S.VLDL.PL, S.VLDL.TG | None |
| LDL | ALDL.C, L.LDL.C, L.LDL.CE, L.LDL.FC, L.LDL.L, L.LDL.P, L.LDL.PL,M.LDL.C, M.LDL.CE, M.LDL.L, M.LDL.P,M.LDL.PL, S.LDL.C, S.LDL.L, S.LDL.P | None |
| HDL | S.HDL.TG, XL.HDL.TG | M.HDL.C, M.HDL.CE |

## UVMR and MVMR

A total of 66 traits were associated with CHD at or below the Bonferroni-corrected level (p = 0.05/97, *Table 1*). Two genetically predicted lipid exposures (M.HDL.C, M.HDL.CE) were negatively associated with CHD and 64 were positively associated (Table 3). In an MVMR model including only the 66 Bonferroni-significant traits, fitted with the purpose of illustrating the instability of IVW-MVMR in conditions of severe collinearity, conditional *F*-statistic (CFS) (Materials and methods) was lower than 2.2 for all exposures (with a mean of 0.81), highlighting the severe weak instrument problem. In *Appendix 1—figure 3*, the MVMR estimates are plotted against the corresponding univariable MR (UVMR) estimates. We interpret the reduction in identified effects as a result of the drop in precision in the MVMR model (variance inflation). Only the independent causal estimate for ApoB reached our pre-defined significance threshold and was less precise ($OR_{MVMR}$ (95% CI): $1.031(1.012, 1.37)$, $OR_{UVMR}$ (95% CI): $1.013(1.01, 1.016)$ (*Appendix 1—figure 4*). We note that, for M.LDL.PL, the UVMR estimate ($1.52(1.35, 1.71)$, $p < 10^{-10}$)) had an opposite sign to the MVMR estimate ($OR_{MVMR} = 0.905(0.818, 1.001)$).

To see if the application of a weak instrument robust MVMR method could improve the analysis, we applied MR GRAPPLE (*Wang et al., 2021*). As the GRAPPLE pipeline suggests, the same three-sample MR design described above is employed. In the external selection GWAS study (GLGC), a total of 148 SNPs surpass the genome-wide significance level for the 97 exposures and were used as instruments. Although the method did not identify any of the exposures as significant at nominal or Bonferroni-adjusted significance level, the strongest association among all exposures is ApoB.

## PCA

Standard PCA with no sparsity constraints was used as a benchmark. PCA estimates a square loadings matrix of coefficients with dimension equal to the number of genetically proxied exposures $K$. The coefficients in the first column define the linear combination of exposures with the largest variability (PC1). Column 2 defines PC2, the linear combination of exposures with the largest variability that is also independent of PC1, and so on. This way, the resulting factors seek to reduce redundant information and project highly correlated SNP-exposure associations to the same PC. In PC1, very low-density lipoprotein (VLDL)- and low-density lipoprotein (LDL)-related traits were the major contributors (*Figure 2a*). ApoB received the 8th largest loading (0.1371, maximum was 0.1403 for cholesterol content in small VLDL) and LDL.C received the 48th largest (0.1147). In PC2, high-density lipoprotein (HDL)-related traits were predominant. The first 18 largest positive loadings are HDL-related and 12 describe either large or extra-large HDL traits. PC3 received its scores mainly from VLDL traits. Six components were deemed significant through the permutation-based approach (*Figure 1*, Materials and methods).

In the second-step IVW regression (step 4 in *Figure 1*), MVMR results are presented. A modest yet precise (OR = $1.002(1.0015, 1.0024)$, $p < 10^{-10}$) association of PC1 with CHD was observed. Conversely, PC3 was marginally significant for CHD at the 5% level (OR = 0.998 (0.998, 0.999), p=0.049). Since $\hat{\gamma}$ has been transformed with linear coefficients (visualised in loadings matrix, *Figure 2*), the underlying causal effects are also transformed and interpreting the magnitude of an effect estimate is not

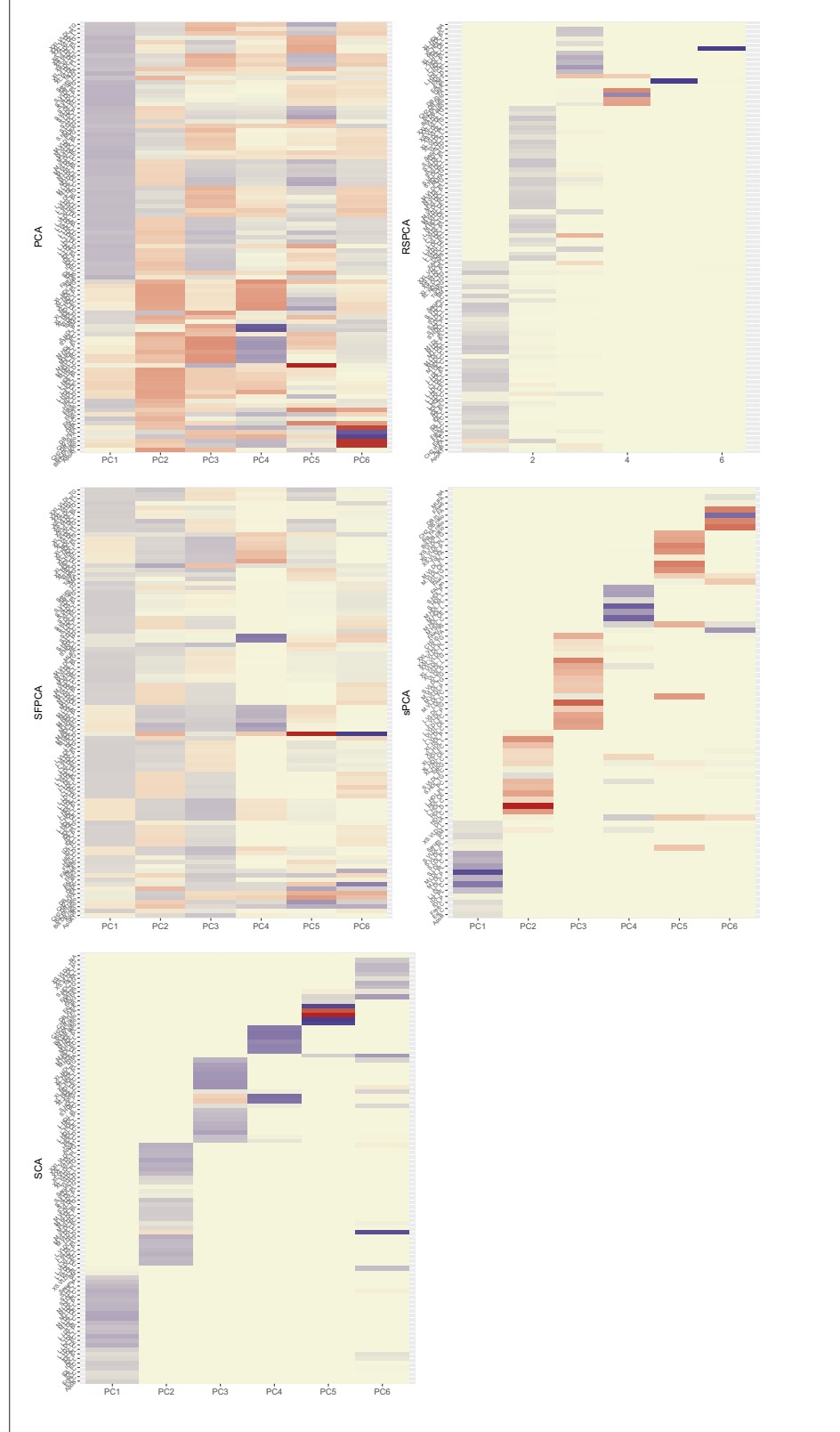

**Figure 2.** Heatmaps for the loadings matrices in the Kettunen dataset for all methods (one with no sparsity constraints [**a**], four with sparsity constraints under different assumptions [**b–e**]). The number of the exposures plotted on the vertical axis is smaller than $K = 97$ as the exposures that do not contribute to any of the sparse principal components (PCs) have been left out. Blue: positive loading; red: negative loading; yellow: zero.

straightforward, since it reflects the effect of changing the PC by one unit on the outcome; however, significance and orientation of effects can be interpreted. When positive loadings are applied to exposures that are positively associated with the outcome, the MR estimate is positive; conversely, if negative loadings are applied, the MR estimate is negative.

## sPCA methods

We next employed multiple sPCA methods (*Table 2*) that each shrink a proportion of loadings to zero. The way this is achieved differs in each method. Their underlying assumptions and details on differences in optimisation are presented in *Table 2* and further described in Materials and methods.

### RSPCA (*Croux et al., 2013*)

Optimisation and the KSS criterion pick six PCs to be informative (*Karlis et al., 2003*). The loadings in *Figure 2* show a VLDL-, LDL-dominant PC1, with some small and medium HDL-related traits. LDL.C and ApoB received the 5th and 40th largest positive loadings. PCs 1 and 6 are positively associated with CHD and PCs 3 and 5 negatively so (*Appendix 1—table 1*).

### SFPCA (*Guo et al., 2010*)

The KSS criterion retains six PCs. The loadings matrix (*Figure 2*) shows the 'fused' loadings with the identical colouring. In the two first PCs, all groups are represented. Both ApoB and LDL.C received the seventh and tenth largest loadings, together with other metabolites (*Figure 2*). PC1 (all groups represented) was positively associated with CHD and PC4 (negative loadings from large HDL traits) negatively so (*Appendix 1—table 1*).

### sPCA (*Zou et al., 2006*)

The number of non-zero metabolites per PC was set at $\frac{148}{97} \sim 16$ (see *Appendix 1—figure 6*). Under this level of sparsity, the permutation-based approach suggested that six sPCs should be retained. Seventy exposures received a zero loading across all components. PC1 is constructed predominantly from LDL traits and is positively associated with CHD, but this does not retain statistical significance at the nominal level in MVMR analysis (*Figure 3*). Only PC4 that is comprised of small and medium HDL traits (*Figure 2b*) appears to exert a negative causal effect on CHD (OR (95% CI): $0.9975(0.9955, 0.9995)$). The other PCs were not associated with CHD (all $p$ values $> 0.05$, *Appendix 1—table 1*).

### SCA (*Chen and Rohe, 2021*)

Six components were retained after a permutation test. In the final model, five metabolites were regularised to zero in all PCs (CH2.DB.ratio, CH2.in.FA, FAw6, S.VLDL.C, S.VLDL.FC, *Figure 2*). Little overlap is noted among the metabolites. PC1 receives loadings from LDL and IDL, and PC2 from VLDL. The contribution of HDL to PCs is split in two, with large and extra-large HDL traits contributing to PC3 and small and medium ones to PC4. PC1 and PC2 were positively associated with CHD (*Appendix 1—table 1*, *Figure 3*). PC4 was negatively associated with CHD.

## Comparison with UVMR

In principle, all PC methods derive independent components. This is strictly the case in standard PCA, where subsequent PCs are perfectly orthogonal, but is only approximately true in sparse implementations. We hypothesised that UVMR and MVMR could provide similar causal estimates of the associations of metabolite PCs with CHD. The results are presented in *Figure 3* and concordance between UVMR and MVMR is quantified with the $R^2$ from a linear regression. The largest agreement of the causal estimates is observed in PCA. In the sparse methods, SCA (*Chen and Rohe, 2021*) and sPCA (*Zou et al., 2006*) provide similarly consistent estimates, whereas some disagreement is observed in the estimate of PC6 for RSPCA (*Croux et al., 2013*) on CHD.

A previous study implicated LDL.c and ApoB as causal for CHD (*Zuber et al., 2020b*). In *Appendix 1—figure 7*, we present the loadings for these two exposures across the PCs for the various methods. Ideally, we would like to see metabolites contributing to a small number of components for the sparse methods. Using a visualisation technique proposed by *Kim and Kim, 2012*, this is indeed observed (see *Appendix 1—figure 7*). In PCA, LDL.c and ApoB contribute to multiple PCs,

**Table 2.** Overview of sparse principal component analysis (sPCA) methods used. KSS: Karlis-Saporta-Spinaki criterion. Package: *R* package implementation; Features: short description of the method; Choice: method of selection of the number of informative components in real data; PCs: number of informative PCs.

| Method | Package | Authors | Features | Choice | PCs |
|---|---|---|---|---|---|
| RSPCA | *pcaPP* | **Croux et al., 2013** | Robust sPCA (RSPCA), different measure of dispersion ($Q_n$) | Permutation KSS | 6 |
| SFPCA | Code in publication, Supplementary Material | **Guo et al., 2010** | Fused penalties for block correlation | KSS | 6 |
| sPCA | *elasticnet* | **Zou et al., 2006** | Formulation of sPCA as a regression problem | KSS | 6 |
| SCA | *SCA* | **Chen and Rohe, 2021** | Rotation of eigen vectors for approximate sparsity | Permutation KSS | 6 |

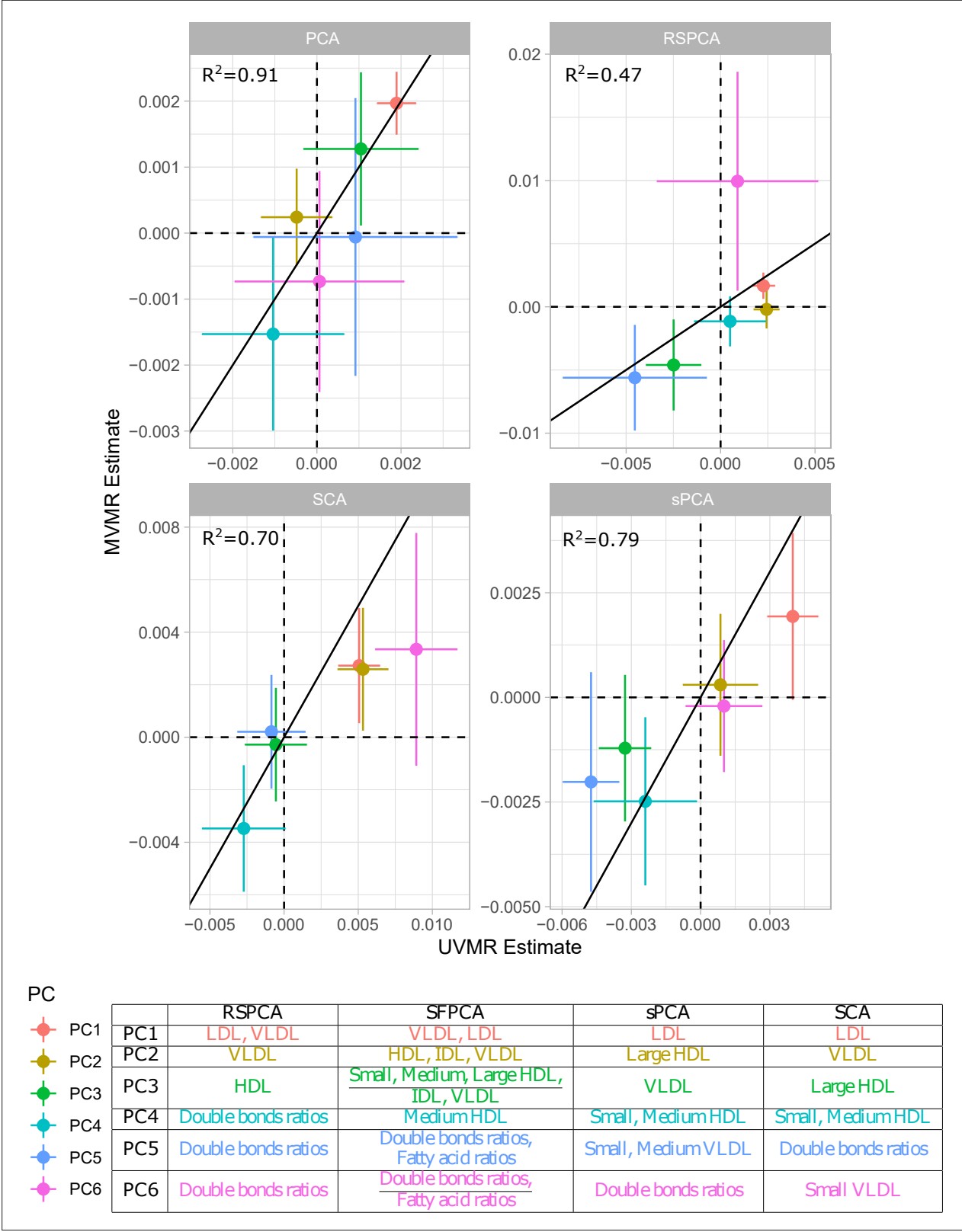

**Figure 3.** Comparison of univariable Mendelian randomisation (UVMR) and multivariable MR (MVMR) estimates and presentation of the major group represented in each principal component (PC) per method.

**Table 3.** Results for principal component analysis (PCA) approaches.

Overlap: Percentage of metabolites receiving non-zero loadings in ≥1 component. Overlap in PC1, PC2: overlap as above but exclusively for the first two components which by definition explain the largest proportion of variance. Very low-density lipoprotein (VLDL), low-density lipoprotein (LDL), and high-density lipoprotein (HDL) significance: results of the IVW regression model with CHD as the outcome for the respective sPCs (the sPCs that mostly received loadings from these groups). The terms VLDL and LDL refer to the respective transformed blocks of correlated exposures; for instance, VLDL refers to the weighted sum of the correlated VLDL-related $\hat{\gamma}$ associations, such as VLDL phospholipid content and VLDL triglyceride content. †: RSPCA projected VLDL- and LDL-related traits to the same PC (sPC1). ‡: SCA discriminated HDL molecules in two sPCs, one for traits of small- and medium-sized molecules and one for large- and extra-large-sized.

|  | PCA | RSPCA | SFPCA | sPCA | SCA |
|---|---|---|---|---|---|
| Overlap | 1 | 0.938 | 1 | 0.187 | 0.196 |
| Overlap in PC1,PC2 | 1 | 0.433 | 1 | 0.010 | 0 |
| Sparse % | 0 | 0.474 | 0.082 | 0.835 | 0.796 |
| VLDL significance in MR† | Yes | No | Yes | No | Yes |
| LDL significance in MR | No | Yes | No | No | Yes |
| HDL significance in MR‡ | Yes | Yes | Yes | No | No |
| Small, medium HDL significance in MR | Yes | No | Yes | Yes | Yes |

whereas the sPCA methods limit this to one PC. Only in RSPCA do these exposures contribute to two PCs. In the second-step IVW meta-analysis, it appears that the PCs comprising of predominantly VLDL/LDL and HDL traits robustly associate with CHD, with differences among methods (*Table 3*).

## Instrument strength

Instrument strength for the chosen PCs was assessed via an $F$-statistic, calculated using a bespoke formula that accounts for the PC process (see Materials and methods and Appendix). The $F$-statistics for all transformed exposures cross the cutoff of 10. There was a trend for the first components being more strongly instrumented in all methods (see *Appendix 1—figure 5*), which is to be expected. In the MVMR analyses, the CFS for all exposures was less than three. Thus the move to PC-based analysis significantly improved instrument strength and mitigated against weak instrument bias.

## Simulation studies

We consider the case of a data generating mechanism that reflects common scenarios found in real-world applications. Specifically, we consider a set of exposures $X$, which can be partitioned into blocks based on shared genetics. Certain groups of variants contribute exclusively to specific blocks of exposures, while having no effect on other blocks. This in turn leads to substantial correlation among the exposure blocks and a much reduced correlation of between exposure blocks, due only to shared confounding. This is visualised in *Figure 4a*. This data structure acts to reduce the instruments' strength in jointly predicting all exposures. The dataset consists of $n$ participants, $k$ exposures, $p$ SNPs (with both $k$ and $p$ consisting of $b$ discrete, equally sized blocks) and a continuous outcome, $Y$. We split the simulation results into one illustrative example (for didactic purposes) and one high-dimensional example.

## Simple illustrative example

We generate data under the mechanism presented in *Figure 4a*. That is, with six individual exposures $X_1, ..., X_6$ split into two distinct blocks ($X_1 - X_3$ and $X_4 - X_6$). A continuous outcome $Y$ is generated that is only causally affected by the exposures in block 1 ($X_1 - X_3$). A range of sample sizes were used in the simulation in order to give a range of CFS values from approximately 2–80. We apply (a) MVMR with the six individual exposures separately, and (b) PCA and SCA. The aim of approach (b) is to demonstrate the impact of reducing the six-dimensional exposure into two PCs, so that the first

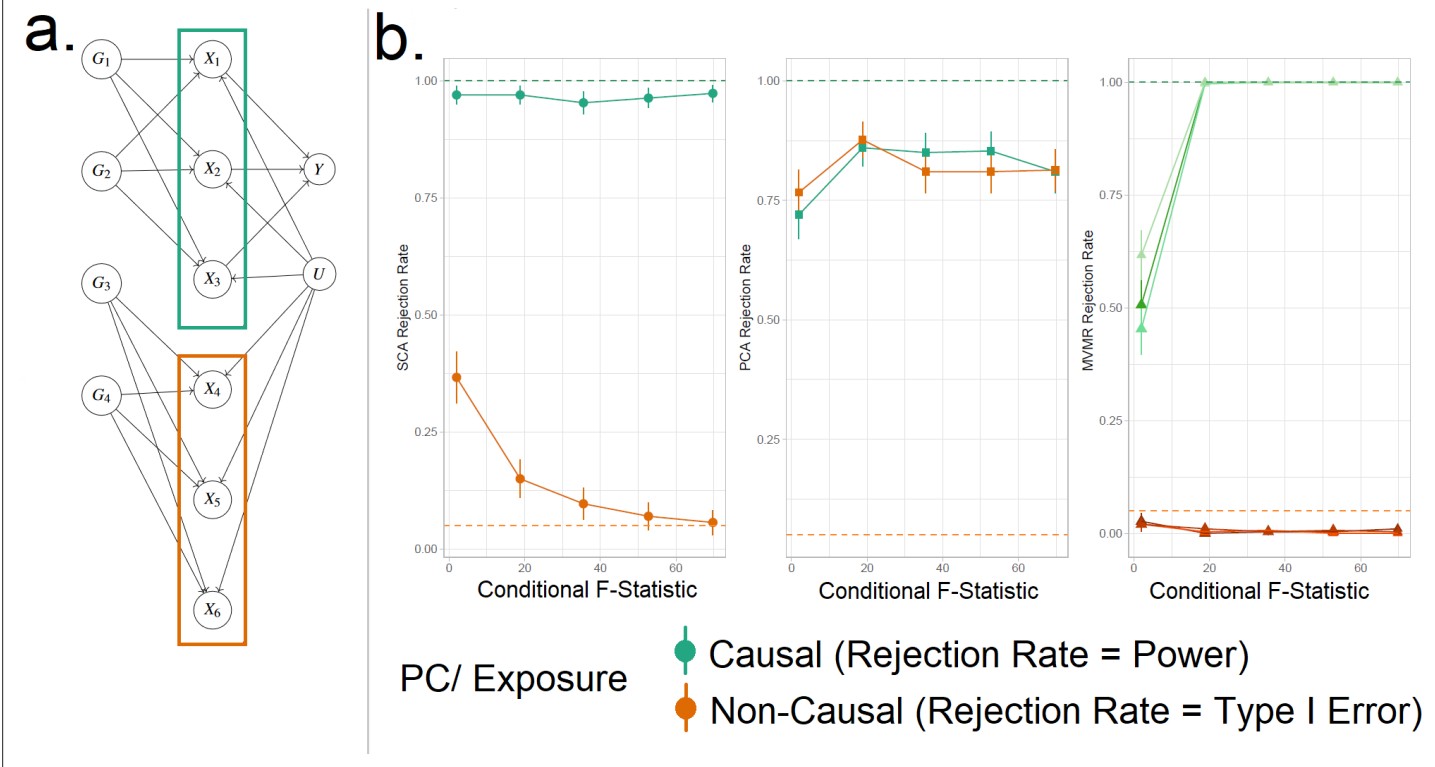

**Figure 4.** Simulation Study Outline. (**a**) Data generating mechanism for the simulation study, illustrative scenario with six exposures and two blocks. In red boxes, the exposures that are correlated due to a shared genetic component are highlighted. (**b**) Simulation results for six exposures and three methods (sparse component analysis [SCA] [***Chen and Rohe, 2021***], principal component analysis [PCA], multivariable Mendelian randomisation [MVMR]). The exposures that contribute to $Y$ ($X_{1-3}$) are presented in shades of green colour and those that do not in shades of red ($X_{4-6}$). In the third panel, each exposure is a line. In the first and second panels, the PCs that correspond to these exposures are presented *as single lines* in green and red. Monte Carlo SEs are visualised as error bars. Rejection rate: proportion of simulations where the null is rejected.

PC has high loadings for block 1 ($X_1 - X_3$) and the second PC has high loadings for block 2 ($X_4 - X_6$). Although two PCs were chosen by both PCA methods using a KSS criterion in a large majority of cases, to simplify the simulation interpretation we fixed a priori the number of PCs at two across all simulations.

Our primary focus was to assess the rejection rates of MVMR versus PCA rather than estimation, as the two approaches are not comparable in this regard. To do this we treat each method as a test, which obtains true positive (TP), true negative (TN), false positive (FP), and false negative (FN) results. In MVMR, a TP is an exposure that is causal in the underlying model *and* whose causal estimate is deemed statistically significant. In the PCA and sPCA methods, this classification is determined with respect to (a) which exposure(s) determine each PC and (b) if the causal estimate of this PC is statistically significant. Exposures are considered to be *major contributors* to a PC if (and only if) their individual PC loading is larger than the average loading. If the causal effect estimate of a PC in the analysis deemed statistically significant, major contributors that are causal and non-causal are counted as TPs and FPs, respectively. TNs and FNs are defined similarly. Type I error therefore corresponds to the FP rate and power corresponds to the TP rate. All statistical tests were conducted at the $\alpha/B = \alpha/2 = 0.025$ level.

SCA, PCA, and MVMR type I error and power are shown in the three panels (left to right) in ***Figure 4b***, respectively. These results suggest an improved power in identifying true causal associations both with PCA and SCA compared with MVMR when the CFS is weak, albeit at the cost of an inflated type I error rate. As sample size and CFS increase, MVMR performs better. For the PC of the second block's null exposures, PCA seems to have a suboptimal type I error control (red in ***Figure 4b***). In this low-dimensional setting, the benefit of PCA therefore appears to be limited.

## Complex high-dimensional example

The aim of the high-dimensional simulation is to estimate the comparative performance of the methods in a wider setting that more closely resembles real data applications. We simulate genetic data and individual level exposure and outcome data for between $K = 30 - 60$ exposures, arranged in $B = 4 - 6$ blocks. The underlying data generating mechanism and the process of evaluating method performance is identical to the illustrative example, but the number of variants, exposures, and the blocks is increased. We amalgamate rejection rate results across all simulations, by calculating sensitivity (SNS) and specificity (SPC) as:

$$SNS = \frac{TP}{TP + FN} \quad SPC = \frac{TN}{TN + FP}, \tag{1}$$

and then compare all methods by their area under the estimated receiver-operating characteristic (ROC) curve (AUC) using the meta-analytical approach of *Reitsma et al., 2005*. Briefly, the Reitsma method performs a bivariate meta-analysis of multiple studies that report both sensitivity and specificity of a diagnostic test, in order to provide a summary ROC curve. A bivariate model is required because sensitivity and specificity estimates are correlated. In our setting the 'studies'

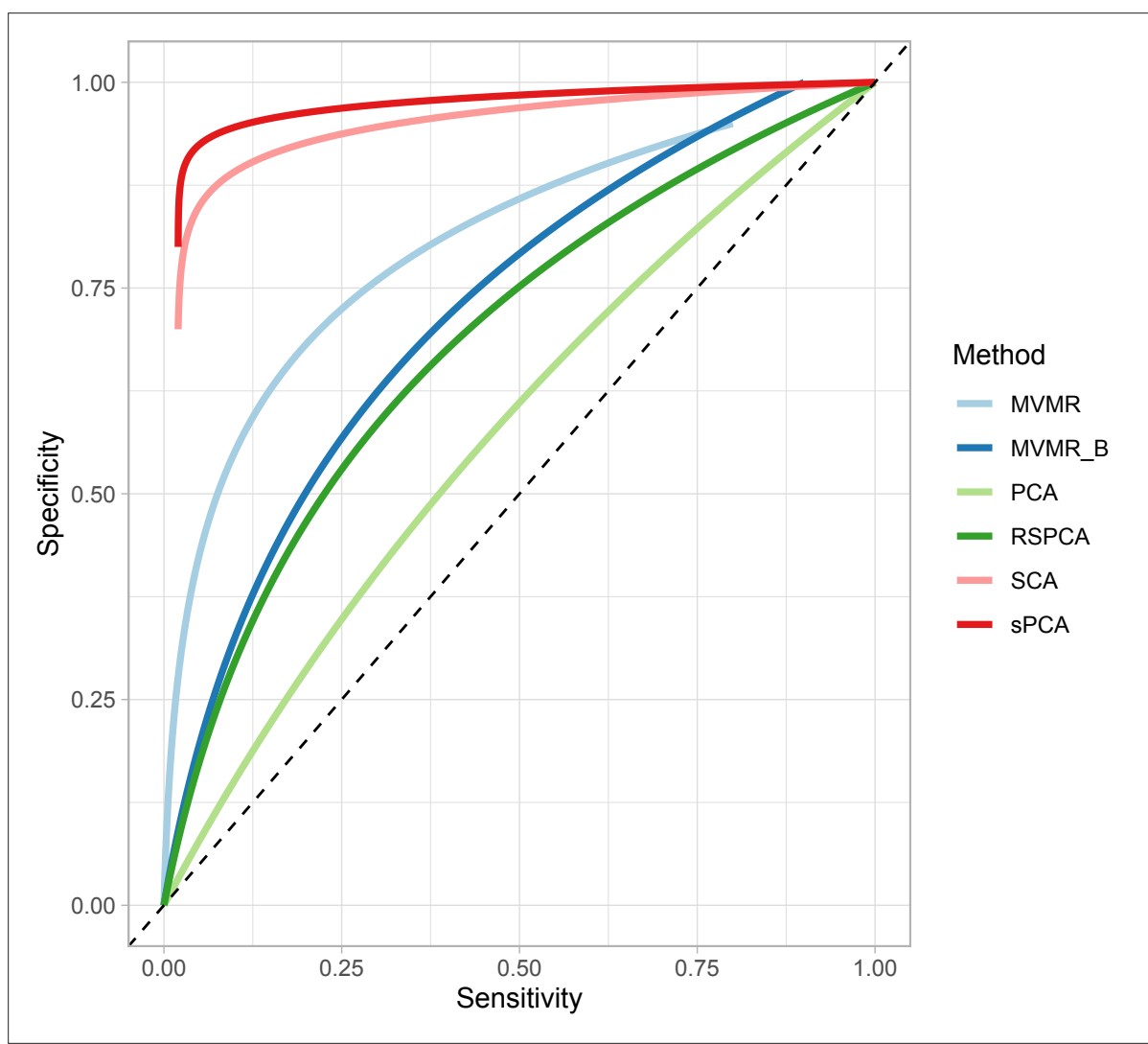

**Figure 5.** Extrapolated receiver-operating characteristic (ROC) curves for all methods. SCA: sparse component analysis (*Chen and Rohe, 2021*) sPCA: sparse PCA (*Zou et al., 2006*) RSPCA: robust sparse PCA (*Croux et al., 2013*); PCA: principal component analysis; MVMR: multivariable Mendelian randomisation; MVMR_B: MVMR with Bonferroni correction.

**Table 4.** Sensitivity and specificity presented as median and interquartile range across all simulations.

Presented as median sensitivity/specificity and interquartile range across all simulations; AUC: area under the receiver-operating characteristic (ROC) curve.

|  | PCA | SCA | sPCA | RSPCA | MVMR_B | MVMR |
|---|---|---|---|---|---|---|
| AUC | 0.56 | 0.919 | 0.941 | 0.644 | 0.660 | 0.712 |
| Sensitivity | 1,0.1 | 1,0.21 | 1, 0.047 | 0.667, 0.251 | 0.222, 0.2 | 0, 0.076 |
| Specificity | 0,0.02 | 0.925,0.772 | 0.936, 0.097 | 0.192, 0.104 | 0.960, 0.048 | 1,0 |
| Youden's J | 0 | 0.584 | 0.778 | −0.061 | 0.192 | 0.044 |

represent the results of different simulation settings with distinct numbers of exposures and blocks. Youden's index $J$ ($J = SNS + SPC − 1$) was also calculated, with high values being indicative of good performance.

Two sPCA methods (SCA [*Chen and Rohe, 2021*], sPCA [*Zou et al., 2006*]) consistently achieve the highest AUC (*Figure 5*). This advantage is mainly driven by an increase in sensitivity for both these methods compared with MVMR. A closer look at the individual simulation results corroborates the discriminatory ability of these two methods, as they consistently achieve high sensitivities (*Appendix 1—figure 10*). Both standard and Bonferroni-corrected MVMR performed poorly in terms of AUC (AUC 0.712 and 0.660, respectively), due to poor sensitivity. PCA performed poorly, with almost equal TP and FP results (AUC 0.560). PCA and RSPCA did not accurately identify negative results (PCA and RSPCA median specificity 0 and 0.192, respectively). This extreme result can be understood by looking at the individual simulation results in *Appendix 1—figure 10*; both PCA and RSPCA cluster to the upper right end of the plot, suggesting a consistently low performance in identifying TN exposures. Specifically, the estimates with both these methods were very precise across simulations and this resulted in many FP results and low specificity. We note a differing performance among the top ranking methods (SCA, sPCA); while both methods are on average similar, the results of SCA are more variable in both sensitivity and specificity (*Table 4*). The Youden's indexes for these methods are also the highest (*Figure 5a*). Varying the sample sizes (mean instrument strength in $\hat{\gamma}$ from $\bar{F} = 221$ to 1109 and mean conditional $F$-statistic $\bar{CFS} = 0.34 − 12.81$) (*Appendix 1—figure 9*) suggests a similar benefit for sparse methods.

Even with large sample sizes ($\bar{F} = 1109.78$, $\bar{CFS} = 12.82$), MVMR can still not discriminate between positive and negative exposures as robustly as the sPCA methods. A major determinant of the accuracy of these methods appears to be the number of truly causal exposures, as in a repeat simulation with only four of the exposures being causal, there was a drop in sensitivity and specificity across all methods. sPCA methods still outperformed other methods in this case, however (*Appendix 1—table 2*).

## What determines PCA performance?

In the hypothetical example of *Figure 4* and indeed any other example, if two PCs are constructed, PCA cannot differentiate between causal and non-causal exposures. The only information used in this stage of the workflow (Steps 2 and 3 in *Figure 1*) is the SNP-$X$ association matrix. Thus, the determinant of projection to common PCs is genetic correlation and correlation due to confounding, rather than how these blocks affect $Y$. Then, if only a few of the exposures truly influence $Y$, it is likely that, PCA will falsely identify the entire block as truly causal. This means the proportion of non-causal exposures within blocks of exposures that truly influence $Y$ is a key determinant of specificity. To test this, we varied the proportion of non-causal exposures by varying the sparsity of the causal effect vector $\beta$ vector and repeated the simulations, keeping the other simulation parameters fixed. As fewer exposures within blocks are truly causal, the performance in identifying TN results drops for SCA (*Appendix 1—figure 12*). However, our simulation still provides a means of making comparisons across methods for a given family of simulated data.

## Discussion

We propose the use of sPCA methods in MVMR in order to reduce high-dimensional exposure data to a lower number of PCs and infer the latter's causal contribution. As the dimensionality of available datasets for MR investigations increases (e.g. in NMR experiments [*Biobank, 2018*] and imaging studies), such approaches are becoming ever more useful. Our results support the notion that sPCA methods retain the information of the initial exposures. Although there is no single optimal method that correctly factorises the SNP-exposure matrix, the goal is to find some grouping of the multiple, correlated exposures such that it may resemble a latent biological variable that generates the data. The SCA (*Chen and Rohe, 2021*) and sPCA (*Zou et al., 2006*) methods performed best in simulation studies and the SCA approach performed best in the positive control example of lipids and CHD. While conventional MR approaches did not identify any protective exposures for CHD, SCA identified a cluster of small and medium HDL exposures that appeared to independently reduce the risk of CHD. This particular subset of HDL particles has previously been implicated in coronary calcification (*Ditah et al., 2016*) and shown to be associated with coronary plaque stability (*Wang et al., 2019*).

By employing sPCA methods in a real dataset (*Kettunen et al., 2016*), we show that the resulting PCs group VLDL, LDL, and HDL traits together, whilst metabolites acting via alternative pathways receive zero loadings. This is a desirable property and indicates that the second-step MR enacted on the PCs obtains causal estimates for intervening on biologically meaningful pathways (*Chipman and Gu, 2005*). This is in contrast with unconstrained PCA, in which all metabolites contribute to all PCs. Previously, *Sulc et al., 2020* used PCA in MR to summarise highly correlated anthropometric variables. To our knowledge, this is the first investigation of different sPCA modalities in the context of MR. Our simulation studies revealed that sPCA methods exhibited superior performance compared to standard PCA, which had high FP rates, and MVMR, which had high FN rates. We additionally provide a number of ways to choose the number of components in a data-driven manner. Our proposed approach of an sPCA method naturally reduces overlap across components; for instance, in a paper by *Sulc et al., 2020*, the authors use PCA and identify four independent axes of variation of body morphology. There are PCs that are driven in common by trunk, arm, and leg lean mass, basal metabolic rate, and BMI; a hypothetical benefit with sparse methods would be reduction of this overlap. This is an important topic for further research. When using PCA without any sparsity constraints, our simulation studies revealed numerous FP results, at the opposite end of the nature of poor performance seen in MVMR; estimates were often misleadingly precise (FN). Although appropriate transformations of the exposures were achieved, we highly recommend exploring additional forms of T1E control to improve the performance of PCA. Nonetheless, sparse methods exhibited superior performance compared to both PCA and MVMR.

A previous work on sparse methods in genetics proposed their usefulness in multi-tissue transcriptome-wide association studies (*Feng et al., 2021*). A finding of the study is that leveraging correlated gene expressions across tissues with sparse canonical correlation analysis improves power to detect gene-trait pairs. Our approach that combines MR with sPCA also showed an improvement in power to detect causal effects of exposures on outcomes.

Our approach is conceptually different from the robust methods that have been developed for standard MVMR in the presence of weak instruments, such as MR GRAPPLE, which attempts to directly adjust point estimates for weak instrument bias, but are not a panacea, especially in the high-dimensional setting discussed here (*Wang et al., 2021*). Furthermore, it reduces the need for a pre-selection of which exposures to include in an MVMR model. We present a complementary workflow through which we can include all available exposures with no prior selection, collate them in uncorrelated and interpretable components, and then investigate the causal contribution of these groups of exposures. It avoids the risk of generating spurious results in such an extreme setting of high collinearity compared with MVMR IVW and MR GRAPPLE formulations. For example, a 2019 three-sample MR study that assessed 82 lipoprotein subfraction risk factors' effects on CHD used an UVMR and a robust extension of MVMR. A positive effect of VLDL- and LDL-related subfractions on CHD was reported, consistent in magnitude across the sizes of the subfractions (*Zhao et al., 2021*). Results were less definitive on the effect of HDL subfractions of varying size on CHD, with both positive and negative effect estimates observed. In our study, the HDL subfractions were uniformly projected to similar subspaces, yielding a single component that was mainly HDL populated in all models, except for the SCA model 15 which projected the small/ medium and large/extra-large HDL traits in two

different components. In all cases, the association of the sPCs with CHD was very low in magnitude. Nevertheless, the direction of effects was in line with the established knowledge on the relationship between lipid classes and CHD.

Within the sPCA methods, there were differences in the results. The sPCA method (*Zou et al., 2006*) favoured a sparser model in which less than 10 metabolites per PC were used. This observation is also made by *Guo et al., 2010*. The SCA method (*Chen and Rohe, 2021*) achieved good separation of the traits and very little overlap was observed. A separation of HDL-related traits according to size, not captured by the other methods, was noted. Clinical relevance of a more high-resolution HDL profiling, with larger HDL molecules mainly associated with worse outcomes, has been previously reported (*Kontush, 2015*).

## Limitations

In the present study, many tuning parameters needed to be set in order to calibrate the PCA methods. We therefore caution against extending our conclusions on the best method outside the confines of our simulation and our specific real data example. Not all available sparse dimensionality reduction approaches were assessed in our investigation and other techniques could have provided better results.

The use of sparsity may raise the concern of neglecting horizontal pleiotropy if a variant influences multiple components, but its weight in a given component is shrunk to zero. This would not occur for standard PCA where no such shrinkage occurs. Currently, our approach is not robust to pleiotropy operating via exposures not included in the model. Our plan is to address this as future work by incorporating median-based MVMR models into the second stage, as done by *Grant and Burgess, 2021*.

### Interpretability

The sPCA approach outlined in this paper enables the user to perform an MVMR analysis with a large number of highly correlated exposures, but one potential downside is that the effect sizes are not as interpretable. Interpreting the causal effects of PCA components (sparse or otherwise) poses a significant challenge. This is because intervening and lowering a specific PC could be actioned by targeting any of the exposures that have a non-zero loading within it. This is in contrast to the causal effect of a single exposure, where the mode of intervention is clear. However, the same ambiguity is often a feature of a real-world intervention, such as a pharmaceutical drug. That is, even if a drug targets a specific lipid fraction, it may exert multiple effects on other lipid fractions that are not pre-planned and are a result of interlinked physiological cascades, overlapping genetics, and interdependent relationships. Identifying such underlying biological mechanisms and pathways is a key step in deciding on the relevance of a PCA derived effect estimate compared to a standard MVMR estimate. We therefore suggest that this approach is best suited for initially testing how large groups of risk factors independently affect health outcomes, before a more focused MVMR within the PCA-identified exposures.

Another limitation of our work is that the instrument selection could have been more precise since we used an external study that investigated total lipid fractions rather than specific size and composition profile characteristics. Future more specific GWAS could improve this, leading to better separation in the genetic predictions of all lipid fractions.

## Conclusion

In the present study, we underline the utility of sparse dimensionality reduction approaches for highly correlated exposures in MR. We present a comprehensive review of methods available to perform dimensionality reduction and describe their performance in a real data application and a simulation.

# Materials and methods
## Approximate relationship to a one-sample analysis

Our approach works directly with summary statistics gleaned from genome-wide association studies that are used to furnish a two-sample analysis, but it can also be motivated from the starting point of a one-sample individual level data. Specifically, assume the data generating model is $\mathbf{y} = \mathbf{X}\beta + \mathbf{u}\mathbf{X} = \mathbf{G}\gamma + \mathbf{V}$ and so that the second-stage model of a two-stage least squares procedure is $\mathbf{y} = \hat{\mathbf{X}}\beta + \tilde{\mathbf{u}}$, where $\hat{\mathbf{X}} = \mathbf{G}\hat{\gamma}$. PCA on $\hat{\mathbf{X}}$ is approximately equivalent to PCA on $\hat{\gamma}$ since $\hat{\mathbf{X}}^T\hat{\mathbf{X}} = \hat{\gamma}^T\hat{\gamma}$ if $\mathbf{G}$ is normalised so

**Table 5.** Two-sample Mendelian randomisation (MR).
Study characteristics.

|  | First author | Year | PMID | *N* | Cases | Controls | Study name (population) |
|---|---|---|---|---|---|---|---|
| Metabolites | Kettunen | 2016 | 27005778 | 24,925 |  |  | NMR GWAS (European) |
| CHD | Nelson | 2017 | 28714975 | 453,595 | 113,937 | 339,658 | CARDIoGRAMplusC4D (European) |

that $\hat{\gamma}$ represents standardised effects. In the appendix we provide further simulation results that show that the loadings matrix derived from a PCA on $\hat{\mathbf{X}}$ and $\hat{\gamma}$ are asymptotically equivalent.

## Data

The risk factor dataset reports the associations of 148 genetic variants (SNPs) with 118 NMR-measured metabolites ((*Table 5*) , *Kettunen et al., 2016*). This study reports a high-resolution profiling of mainly lipid subfractions. To focus on lipid-related traits, we exclude amino acids and retain 97 exposures for further analyses. Fourteen size/density classes of lipoprotein particles (ranging from extra small [XS] HDL to extra-extra-large [XXL] VLDL) were available and, for each class, the traits of total cholesterol, triglycerides, phospholipids, and cholesterol esters content, and the average diameter of the particles were additionally provided. Through the same procedure, estimates from NMR on genetic associations of amino acids, apolipoproteins, fatty and fluid acids, ketone bodies, and glycerides were also included. Instrument selection for this dataset has been previously described (*Zuber et al., 2020a*). Namely, 185 variants were selected, based on association with either one of: LDL-cholesterol, HDL-cholesterol, or triglycerides in the external sample of the Global Lipid Genetics Consortium at the genome-wide level ($p < 5 \times 10^{-8}$) (*Do et al., 2013*). Then, this set was pruned to avoid inclusion of SNPs in linkage disequilibrium (LD) (threshold: $r^2 < 0.05$) and filtered (variants in distance less than 1 megabase pairs were excluded) resulting in the final set. This pre-processing strategy was performed with a view to study CHD risk.

Positive control outcome assessment is recommended in MR as an approach of investigating a risk factor that has an established causal effect on the outcome of interest (*Burgess et al., 2020*). We used CHD as a positive control outcome, given that lipid fractions are known to modulate its risk, with VLDL- and LDL-related traits being positively associated with CHD and HDL-related traits negatively (*Burgess et al., 2020*). Summary estimates from the CARDIoGRAMplusC4D Consortium and UK Biobank meta-analysis (*Nelson et al., 2017*; *Deloukas et al., 2013*) were used. For both datasets, a predominantly European population was included, as otherwise spurious results could have arisen due to population-specific, as opposed to CHD-specific, differences (*Vilhjálmsson and Nordborg, 2013*).

## MR assumptions

The first assumption (IV1) states that IVs should be associated with the exposure. The second assumption (IV2) states that the IVs should be independent of all confounders of the risk factor-outcome

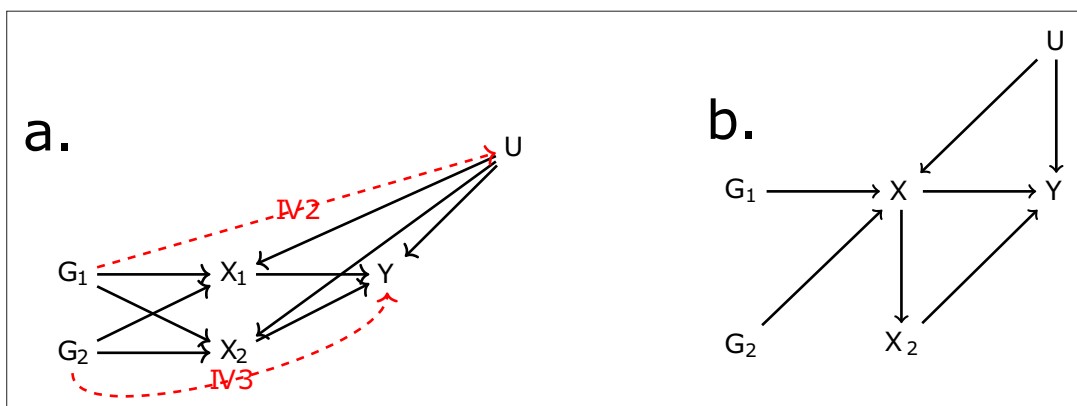

**Figure 6.** Directed acyclic graph (DAG) for the multivariable Mendelian randomisation (MVMR) assumptions. IV2, IV3: instrumental variable assumptions 2 and 3.

association (IV2) and, finally, independent of the outcome conditional on the exposure and the confounders (IV3). The validity of the final two assumptions cannot be directly verified with the available data. For the inferences to be valid, it is necessary that the three IV assumptions apply (**Davies et al., 2018**). These assumptions are illustrated for the case of two exposures in **Figure 6a**. In situations when IV3 is not deemed likely, additional risk factors that are possible mediators can be introduced in an MVMR model (**Burgess and Harshfield, 2016**). Additional assumptions should hold for MVMR results to be valid. Particularly, (a) all included exposures have to be associated with at least one of the IVs, and (b) there should be no evidence of multicollinearity. If there is significant overlap, for example if $G_1$ and $G_2$ are associated with both exposures $X_1$ and $X_2$, but only with $X_2$ through $X_1$ as in **Figure 6b**, this may result in conditionally weak instruments (**Sanderson et al., 2020**). The latter assumption and the way it limits eligibility of high-dimensional, correlated exposures is a key motivation for the present work.

## UVMR and MVMR

To examine how each of the metabolites is causally associated with CHD, a UVMR approach was first undertaken under the two-sample summary data framework. This uses SNP-exposure and SNP-outcome GWAS summary statistics ($\hat{\gamma}$ and $\hat{\Gamma}$, respectively), obtained by regressing the exposure or outcome trait on each SNP individually. Usually, an additional adjustment has been made for variables such as age, gender, and genetic PCs. An UVMR analysis is the most straightforward way to obtain an estimate for the causal effect of the exposure, but is only reliable if all SNPs are valid instruments. However, in the Kettunen dataset, where the exposure set comprises 97 highly correlated lipid fractions, one variant may influence multiple exposures. This will generally invalidate an UVMR analysis and an MVMR approach is more appropriate.

Estimating **Equation 2** provides a general means for representing the mechanics of an UVMR or MVMR analysis:

$$Q_A = \sum_{j=1}^{p}(\frac{1}{\sigma_{Aj}^2})(\hat{\Gamma}_j - \sum_{k=1}^{K}\beta_k\hat{\gamma}_{kj})^2, \quad \text{where } \sigma_{Aj}^2 = \sigma_{Yj}^2 + \sum_{k=1}^{K}\beta_k^2\sigma_{kj}^2 + \sum_{k=1}^{K}\sum_{l=1}^{K}2\beta_l\beta_k\sigma_{lkj}, \quad (2)$$

where

- $\hat{\gamma}_{kj}$ represents the association of SNP $j$ with exposure $k$, with variance $\sigma_{kj}^2$;
- $\hat{\Gamma}_j$ represents the association of SNP $j$ with the outcome, with variance $\sigma_{Yj}^2$;
- $\beta_k$ represents the causal effect of exposure $k$ on the outcome to be estimated.
- $\sigma_{lkj}$ represents $\text{Cov}(\hat{\gamma}_{lj}, \hat{\gamma}_{kj})$. If individual-level data is not available to obtain this quantity, it can be estimated using the intercept from LD score regression (**Bulik-Sullivan et al., 2015**).

In an UVMR analysis there is only one exposure, so that $K=1$, whereas in an MVMR analysis $K \geq 2$ (or in the case of the Kettunen data, $K=97$). In an IVW UVMR or MVMR analysis, the causal effect parameters estimated are obtained by finding the values $\hat{\beta}_1, ..., \hat{\beta}_K$ that minimise $Q_A$, under the simplifying assumption that $\sigma_A^2 = \sigma_{Yj}^2$. This is justified if all causal effects are zero, or if the uncertainty in the SNP-exposure associations is negligible (the no measurement error [NOME] assumption). In the general MVMR context, the NOME assumption is approximately satisfied if the CFS for each exposure are large, but if it is not, then IVW MVMR estimates will suffer from weak instrument bias. For exposure $k$, $CFS_k$ takes the form

$$CFS_k = \frac{Q_{X_k}}{p - (K - 1)},$$

where:

$$Q_{Xk} = \sum_{j=1}^{p}(\frac{1}{\sigma_{Xkj}^2})(\hat{\gamma}_{kj} - \sum_{m\neq k}\hat{\delta}_m\hat{\gamma}_{mj})^2, \quad \text{where } \sigma_{Xkj}^2 = \sigma_{Xkj}^2 + \sum_{m\neq k}\hat{\delta}_m^2\sigma_{mj}^2 + \sum_{m\neq k}\sum_{l\neq k}2\hat{\delta}_m\hat{\delta}_l\sigma_{mlj}, \quad (3)$$

where $\hat{\delta}$ is estimated by regressing an exposure $X_k$ on all other exposures $X_{-k}$ by OLS. If an exposure $k$ can be accurately predicted conditionally on other exposures, then there won't be sufficient

variation or 'good heterogeneity' (*Sanderson et al., 2019*) in $Q_{Xk}$ and CFS will generally be small. This will be the case whenever there is a high degree of correlation between the SNP-exposure association estimates, for example as in an MVMR analysis of all 118 lipid fractions in the Kettunen data. One option to address this would be to use weak instrument robust MVMR, such as MR-GRAPPLE (*Wang et al., 2021*). This method performs estimation using the full definition of $\sigma_A^2$ in *Equation 2* and using a robust loss function that additionally penalises larger outliers in the data. It can work well for MVMR analyses of relatively small numbers of exposures (e.g. up to 10) and relatively weak instruments (CFS as low as 3), but the dimension and high correlation of the Kettunen data is arguably too challenging. This motivates the use of dimension reduction techniques, such as PCA.

## Dimension reduction via PCA

PCA is a singular value decomposition a given matrix of the $p \times K$ matrix of SNP-exposure associations $\hat{\gamma}$ as:

$$\hat{\gamma} = UDV^T$$

where $U$ and $V$ are orthogonal matrices and $D$ is a square matrix whose diagonal values are the variances explained by each component and all off-diagonal values are 0. $V$ is the loadings matrix and serves as an indicator of the contribution of each metabolite to the transformed space of the PCs. The matrix $UD$ (PCs/scores matrix) is used in the second-step IVW regression in place of $\hat{\gamma}$. As $V$ estimation does not aim for sparsity, all exposures will contribute to some degree to all components, making the interpretation more complicated. Therefore, we assessed multiple sPCA methods that intentionally limit this.

## sPCA (Zou et al.)

sPCA by *Zou et al., 2006*, estimates the loadings matrix through an iterative procedure that progressively penalises exposures so that they do not contribute to certain PCs. In principle, this leads to a more clear picture for the consistency of each PC. This is performed as follows:

1. Setting a fixed matrix, the following elastic net problem is solved $\xi_j = argmin_\xi(\alpha_j - \xi)^T\hat{\gamma}^T\hat{\gamma}(\alpha_j - \xi) + \lambda_{1j}\|\xi\| + \lambda\|\xi\|^2$, where $j$ is the PC.
2. For a fixed $\Xi$, $\hat{\gamma}^T\hat{\gamma}\Xi = UDV^T$ is estimated and update $A = UV^T$.
3. Repeat steps 1 and 2 until convergence.

Here, $\lambda_1$ is an $L1$ sparsity parameter that induces sparsity, $\lambda_2$ is an $L2$ parameter that offers numerical stability, and $\Xi$ is a matrix with sparsity constraints for each exposure (*Zou and Xue, 2018*). As a result of the additional $\lambda_1\|\xi\|$ norm, there is sparsity in the loadings matrix and only some of the SNP-exposure associations $\hat{\gamma}$ contribute to each PC, specifically a particular subset of highly correlated exposures in $\hat{\gamma}$.

## RSPCA

This approach differs in that it employs a robust measure of dispersion that is not unduly influenced by large single values of $\hat{\gamma}$ that contribute a large amount to the total variance (*Rousseeuw and Croux, 1993*; *Heckert, 2003*). As above, an $L_1$ norm is used to induce sparsity. For optimisation, the tradeoff product optimisation is maximised. It does not impose a single $\lambda$ value on all PCs, thus allowing different degrees of sparsity.

## SFPCA (*Guo et al., 2010*)

A method that can in theory exploit distinct correlation structures. Its goal is to derive a loadings matrix in which highly positively correlated variables are similar in sign and highly negative ones are opposite. Similar magnitudes also tend to be obtained for those variables that are in the same blocks in the correlation matrix. Like the sPCA optimisation in *Zou et al., 2006*, sparse fused PCA (SFPCA) works by assigning highly correlated variables the exact same loadings as opposed to numerically similar ones (*Figure 2d*). This is achieved with two norms in the objective function: $\lambda_1$ which regulates the $L_1$ norm that induces sparsity and $\lambda_2$ for the $L2$ regularisation (squared magnitude of $\hat{\gamma}$) to guard

against singular solutions. A grid search is used to identify appropriate parameters for $\lambda_1$ and $\lambda_2$. The following criterion is used:

$$min_{A,\Xi}\|\hat{\gamma} - \hat{\gamma}\Xi A^T\|_F + \lambda_1\|\xi\| + \lambda_2|\rho_{s,t}|\|\xi_{s,t} - sign(\rho_{s,t}\xi_{t,k})|,$$

such that $A^T A = I_K$. The 'fused' penalty (last term) purposely penalises discordant loadings for variables that are highly correlated. The choice of the sparsity parameters is based on a Bayesian information criterion (BIC).

## SCA

SCA (*Chen and Rohe, 2021*) is motivated by the relative inadequacy of the classic approaches in promoting significant sparsity. It addresses this by rotating the eigenvectors to achieve approximate sparsity whilst keeping the proportion of variance explained the same. Simulation studies show the technique works especially well in high-dimensional settings such as gene expression data, among other examples (*Chen and Rohe, 2021*).

## Choice of components

In all dimensionality reduction approaches applied to correlated variables, there is no upper limit to how many transformed factors can be estimated. However, only a proportion of them are likely to be informative in the sense of collectively explaining a meaningful amount of total variance in the original dataset. To guide this choice, a permutation-based approach was implemented (*Coombes and Wang, 2019*) as follows: Firstly, the $\hat{\gamma}$ matrix was randomly permuted and the sPCA method of interest was applied on the permuted set. The computed eigenvalues are assumed to come from a null distribution consistent with a non-informative component. This process is repeated multiple times (e.g. $perm = 1000$) and the mean eigenvalues for all components stored. Finally, the sPCA method is performed in the original $\hat{\gamma}$ matrix and whichever component has an eigenvalue larger than the mean of the permuted sets is considered informative and kept. Due to the computational complexity of the permutation method, particularly for SFPCA, an alternate method - the KSS criterion (*Karlis et al., 2003*) - was also used. This is based on a simple correction on the minimum non-trivial eigenvalue (Cutoff$_{KSS} = 1 + 2\sqrt{\frac{K-1}{p-1}}$). The authors show that the method is robust to non-normal distributions (*Karlis et al., 2003*). Although KSS was not compared with the above-described permutation approach, it performed better than simpler approaches, such as choosing those PCs whose eigenvalue is larger than 1 (Kaiser criterion), the broken stick method (*Jolliffe, 2002*) and the Velicer method (*Velicer, 1976*).

## Instrument strength of PCs

In MVMR, the $IV1$ assumption requires a set of genetic variants that robustly associate with at least one of the exposures $X$ (*MR assumptions*). This is quantified by CFS in *Equation 2* (*Sanderson et al., 2020*). With summary statistics of the SNP-$X$ associations $\hat{\gamma}_{p,k}$ ($p$: SNP, $k$: exposure), the mean $F$-statistic for exposure $k$ used in a standard UVMR analysis is the far simpler expression

$$F_k = \frac{\sum_{j=1}^{p}(\frac{\hat{\gamma}_{j,k}}{SE_{\hat{\gamma}_{j,k}}})^2}{p} \tag{4}$$

We provide a dedicated formula for estimating instrument strength measures for the $F$-statistic for the PCs that is closely related to *Equation 3* rather than *Equation 2*. This simplification is due to the fact that an MVMR analysis of a set of PCs is essentially equivalent to an UMVR analysis of each exposure separately. The full derivation is reported in the section 'Instrument strength' of the Appendix.

## Acknowledgements

We would like to thank Dr. Stephen Burgess and two anonymous reviewers that provided constructive comments on the extension of the approach to individual-level data, on improving and clarifying the applied analyses, and on updating and communicating the simulation results.

## Additional information

### Competing interests
Dipender Gill: is a part-time employee of Novo Nordisk. The other authors declare that no competing interests exist.

### Funding

| Funder | Grant reference number | Author |
|---|---|---|
| State Scholarships Foundation | | Vasileios Karageorgiou |
| Expanding Excellence in England | | Vasileios Karageorgiou |

The funders had no role in study design, data collection and interpretation, or the decision to submit the work for publication.

### Author contributions
Vasileios Karageorgiou, Data curation, Software, Formal analysis, Investigation, Methodology, Writing - original draft, Writing – review and editing; Dipender Gill, Conceptualization, Data curation, Supervision, Investigation, Methodology, Writing – review and editing; Jack Bowden, Software, Formal analysis, Supervision, Investigation, Visualization, Methodology, Writing – review and editing; Verena Zuber, Conceptualization, Data curation, Software, Formal analysis, Supervision, Investigation, Visualization, Methodology, Writing – review and editing

### Author ORCIDs
Vasileios Karageorgiou http://orcid.org/0000-0002-7173-9967

### Decision letter and Author response
Decision letter https://doi.org/10.7554/eLife.80063.sa1
Author response https://doi.org/10.7554/eLife.80063.sa2

## Additional files

### Supplementary files
• MDAR checklist

### Data availability
The GWAS summary statistics for the metabolites (http://www.computationalmedicine.fi/data/NMR_GWAS/) and CHD (http://www.cardiogramplusc4d.org/) are publicly available. We provide code for the SCA function, the simulation study and related documentation on GitHub (https://github.com/vaskarageorg/SCA_MR/, copy archived at *Karageorgiou, 2023*).

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

## Appendix 1

### Instrument strength

Since we transform $\hat{\gamma}$ and obtain a matrix of lower dimensionality, formula (*Equation 4*) can't be used as there is no longer a one-to-one correspondence of the $SE_s$ with the PCs. Likewise, a conditional *F*-statistic for the PCs also cannot be computed for this reason. We aim to arrive at a modified formula that bypasses this issue. For this purpose, we take advantage of two concepts, first an expression of the *F*-statistic for an exposure $k$ ($F_k$) in matrix notation and, second, the use of this expression to estimate *F*-statistics for the PCs ($F_{PC}$) from $\hat{\gamma}$ decomposition.

We make the assumption that the uncertainty in the $\hat{\gamma}_{G,X_K}$ estimates is similar in all $K$ exposures, that is $\hat{\gamma}_{G,X}$ uncertainty estimates do not substantially differ among exposures. This is not implausible as the uncertainty is predominantly driven by sample size and minor allele frequency (*Giambartolomei et al., 2014*). Specifically, the authors of *Giambartolomei et al., 2014*, show that

$$Var(\hat{\gamma}_{X_k}) = \frac{1}{n_k Var(X_k)MAF(1-MAF)},$$

where MAF is the minor allele frequency, $n_k$ is the sample size in exposure $k$, and $Var(X_k)$ is the phenotypic variance. What this means is that, in experiments such as *Kettunen et al., 2016*, where $n_k$ is the same across all exposures and $Var(X_k)$ can be standardised to 1, the main driver of differences in $Var(\hat{\gamma}_{X_k})$ is differences in MAF. As MAF is the same for each SNP across all exposures, the collation of SEs across exposures per SNP is well motivated.

We can then define a matrix $\Sigma$ as follows.

$$\Sigma = \begin{bmatrix} \bar{SE}_1^2 & 0 & 0 & \dots & 0 \\ 0 & \bar{SE}_2^2 & 0 & \dots & 0 \\ \vdots & \vdots & \vdots & \ddots & \vdots \\ 0 & 0 & 0 & \dots & \bar{SE}_p^2 \end{bmatrix}, \; \bar{SE}_j^2 = \frac{\sum_{k=1}^{K} SE_{j,k}^2}{K}$$

The elements in the diagonal represent the mean variance of $\hat{\gamma}$ for each SNP and all off-diagonal elements are zero. What is achieved through this is a summary of the uncertainty in the SNP-$X$ associations that is not sensitive on the dimensions of the exposures. Instead of *Equation 4*, we can then express the vector of the mean *F*-statistics for each exposure $F_{1-K} = [F_1, F_2, ..., F_K]$ as

$$\underset{K \times 1}{F_{1-K}} = \frac{1}{p} \times diag[\; \underset{K \times p}{\hat{\gamma}^T} \times \underset{p \times p}{\Sigma^{-1}} \times \underset{p \times K}{\hat{\gamma}} \;] \tag{5}$$

where $\hat{\gamma}$ is the *matrix* of the SNP-exposure associations. In a simulation study, we generate data under the mechanism in *Appendix 1—figure 1a*. The strength of association is different in the three exposures. It is observed that the estimates with both methods (*Equation 5* and *Equation 4*) align well (*Appendix 1—figure 1b*), supporting the equivalence of the two formulae.

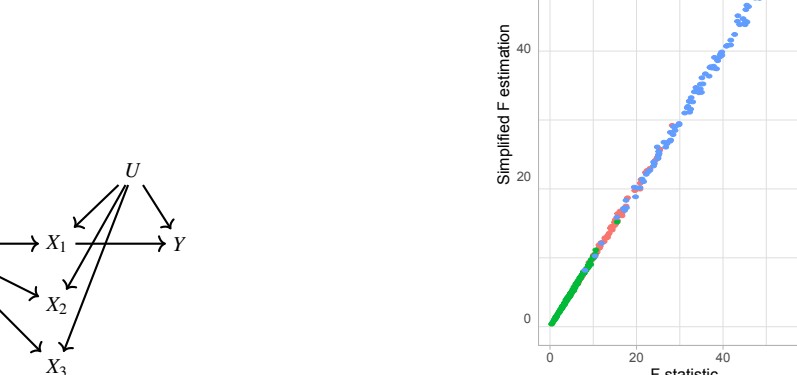

**Appendix 1—figure 1.** Simplified F-statistic Estimation. (**a**) Data generating mechanism. Three exposures with different degrees of strength of association with $G$ are generated $\gamma_1 = 1, \gamma_2 = 0.5, \gamma_3 = 0.1$. (**b**) $F$-statistic for the three exposures $X_1, X_2, X_3$ as estimated by the formulae in *Equation 5* (horizontal axis) and *Equation 4* (vertical axis).

Our second aim is to use this matrix notation formula for the *F*-statistic to quantify the instrument strength of each PC with the respective *F*-statistic ($F_{PC}$). As presented above, we are not limited by the dimensions of point estimates and uncertainty matching exactly and we can use the formula in *Equation 5* and substitute $\hat{\gamma}$ with the PCs. For the PCA approach, where we decompose $\hat{\gamma}$ as $\hat{\gamma} = UDV^T$ and carry forward $M << K$ non-trivial PCs, we use the matrix $UD$ in place of $\hat{\gamma}$. Then, the mean $F_{PC}$ can be estimated as follows.

$$\underset{M \times 1}{F_{PC_{1-M}}} = \frac{1}{p} \times diag[\underset{M \times p}{UD^T} \times \underset{p \times p}{\Sigma^{-1}} \times \underset{p \times M}{UD}] \tag{6}$$

The vector $F_{PC_{1-M}} = [F_{PC_1}, F_{PC_2}, ..., F_{PC_M}]$ contains the $F_{PC}$ statistics for the $M$ PCs. In a similar manner, we estimate $F_{PC}$ for the sPCA methods but, instead of the scores matrix $UD$, we use the scores of the sparse methods. We illustrate the performance of this approach in a simulation study with an identical configuration for exposure generation as the one presented in *Figure 5*. In a configuration with $b = 6$ blocks of $p = 30$ genetically correlated exposures (*Figure 4*), we vary the strength of association $\gamma$ per block. This way, the first block has the highest strength of association and the last block the lowest, quantified by a lower mean *F*-statistic in the exposures of this block (red, *Appendix 1—figure 2*). The instrument strength of the PCs and the sPCs follow closely the corresponding *F*-statistics of the individual exposures; in other words, in a PC of five exposures, $F_{PC_1}$, $F_{SCA_1}$, and $F_{1-5}$ align well (*Appendix 1—figure 2*).

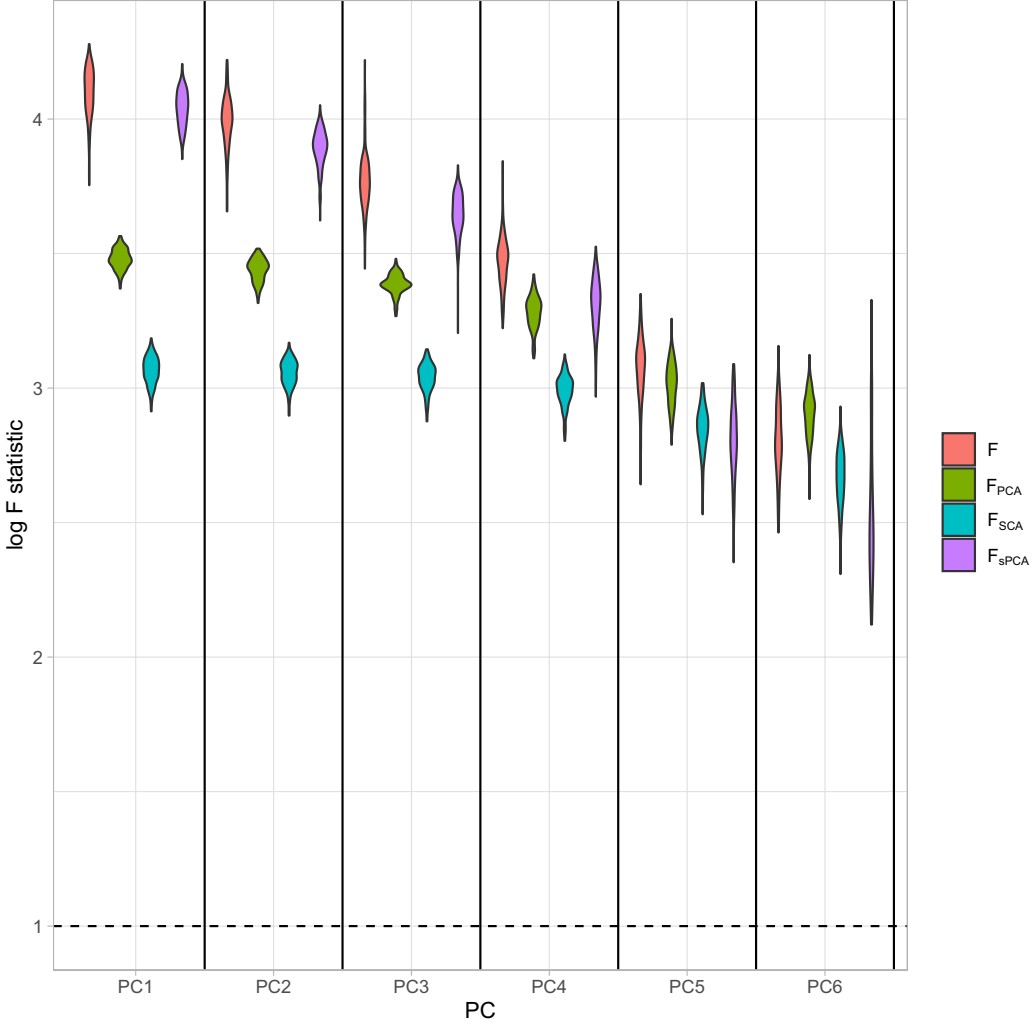

**Appendix 1—figure 2.** Distributions of the *F*-statistics in principal component analysis (PCA) methods and individual (not transformed) exposures. Exposure data in different blocks are simulated with a decreasing strength of association and the correlated blocks map to principal components (PCs). Each distribution represents the *F*-statistics for each PC. In the case of the individual exposures (red), the distributions represent the *F*-statistics for the corresponding exposures. Individual: individual exposures without any transformation; PCA: *F*-statistics for PCA; SCA: sparse component analysis (***Chen and Rohe, 2021***) sPCA: sparse PCA as described by ***Zou et al., 2006***.

## MVMR and UVMR

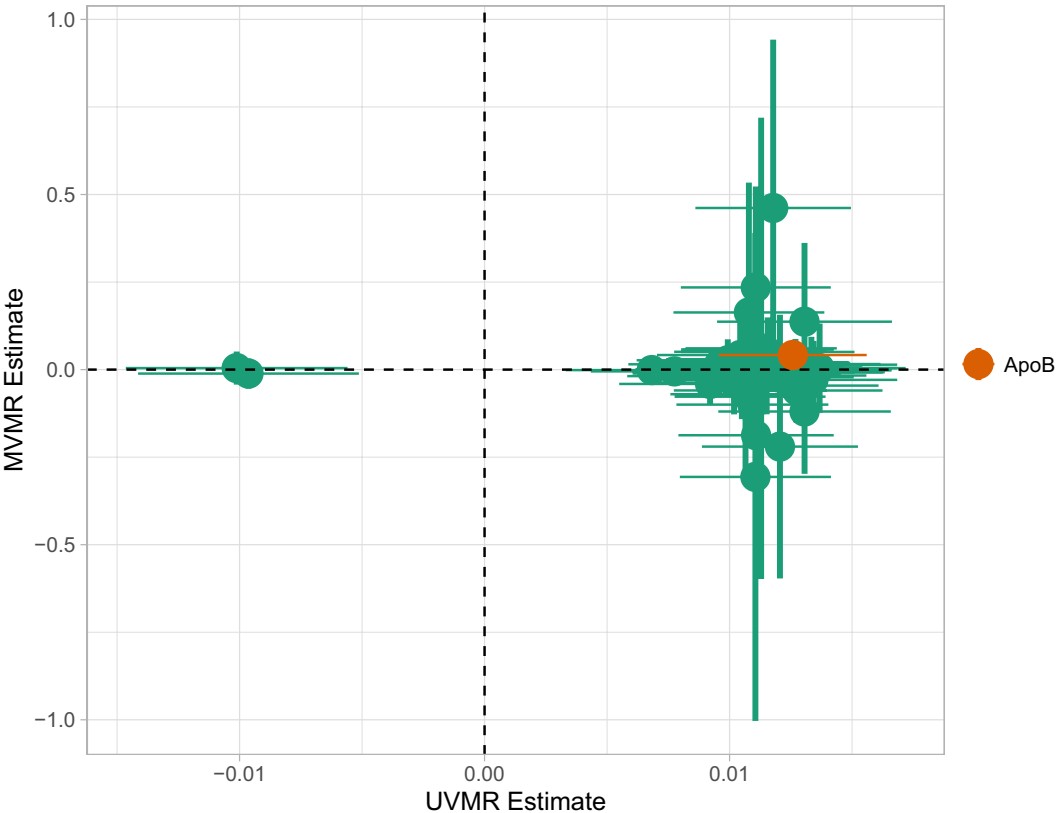

**Appendix 1—figure 3.** Multivariable Mendelian randomisation (MVMR) and univariable MR (UVMR) estimates. Only ApoB is strongly associated with coronary heart disease (CHD). All SEs are larger in the MVMR model (range of $\frac{SE_{MVMR}}{SE_{UVMR}}$).

## MVMR with PC scores

**Appendix 1—table 1.** Estimated causal effects of principal components (PCs) on coronary heart disease (CHD) risk.

PCA: principal component analysis; SCA: sparse component analysis; sPCA: sparse PCA (*Zou et al., 2006*); RSPCA: robust sparse PCA.

| PC | Method | OR | LCI | UCI |
|----|--------|------|--------|--------|
| PC1 | PCA | 1.002 | 1.0015 | 1.0024 |
| PC2 | PCA | 1.0002 | 0.9995 | 1.001 |
| PC3 | PCA | 1.0013 | 1.0001 | 1.0024 |
| PC4 | PCA | 0.9985 | 0.997 | 0.9999 |
| PC5 | PCA | 0.9999 | 0.9978 | 1.002 |
| PC6 | PCA | 0.9993 | 0.9976 | 1.0009 |
| PC1 | SCA | 1.0027 | 1.0005 | 1.0049 |
| PC2 | SCA | 1.0027 | 1.0004 | 1.005 |
| PC3 | SCA | 0.9997 | 0.9976 | 1.0019 |
| PC4 | SCA | 0.9965 | 0.9941 | 0.9989 |

*Appendix 1—table 1 Continued on next page*

*Appendix 1—table 1 Continued*

| PC | Method | OR | LCI | UCI |
|---|---|---|---|---|
| PC5 | SCA | 1.0002 | 0.998 | 1.0024 |
| PC6 | SCA | 1.0034 | 0.9989 | 1.0078 |
| PC1 | sPCA | 1.0019 | 0.9999 | 1.0039 |
| PC2 | sPCA | 1.0003 | 0.9986 | 1.002 |
| PC3 | sPCA | 0.9988 | 0.997 | 1.0005 |
| PC4 | sPCA | 0.9975 | 0.9955 | 0.9995 |
| PC5 | sPCA | 0.998 | 0.9954 | 1.0006 |
| PC6 | sPCA | 0.9998 | 0.9982 | 1.0014 |
| PC1 | RSPCA | 1.0017 | 1.0006 | 1.0027 |
| PC2 | RSPCA | 0.9998 | 0.9983 | 1.0013 |
| PC3 | RSPCA | 0.9954 | 0.9918 | 0.999 |
| PC4 | RSPCA | 0.9989 | 0.9969 | 1.0008 |
| PC5 | RSPCA | 0.9944 | 0.9903 | 0.9986 |
| PC6 | RSPCA | 1.01 | 1.0013 | 1.0188 |
| PC1 | SFPCA | 1.002 | 1.0015 | 1.0025 |
| PC2 | SFPCA | 0.9991 | 0.9979 | 1.0004 |
| PC3 | SFPCA | 0.9998 | 0.9991 | 1.0006 |
| PC4 | SFPCA | 0.9982 | 0.9967 | 0.9997 |
| PC5 | SFPCA | 1.0001 | 0.9977 | 1.0025 |
| PC6 | SFPCA | 1.0009 | 0.9985 | 1.0033 |

## MVMR with IVW and GRAPPLE

A small negative effect for M.LDL.PL is noted as nominally significant in *Appendix 1—figure 4*. This is not concordant with the UVMR direction of effect. In GRAPPLE, no traits surpass the nominal significance threshold.

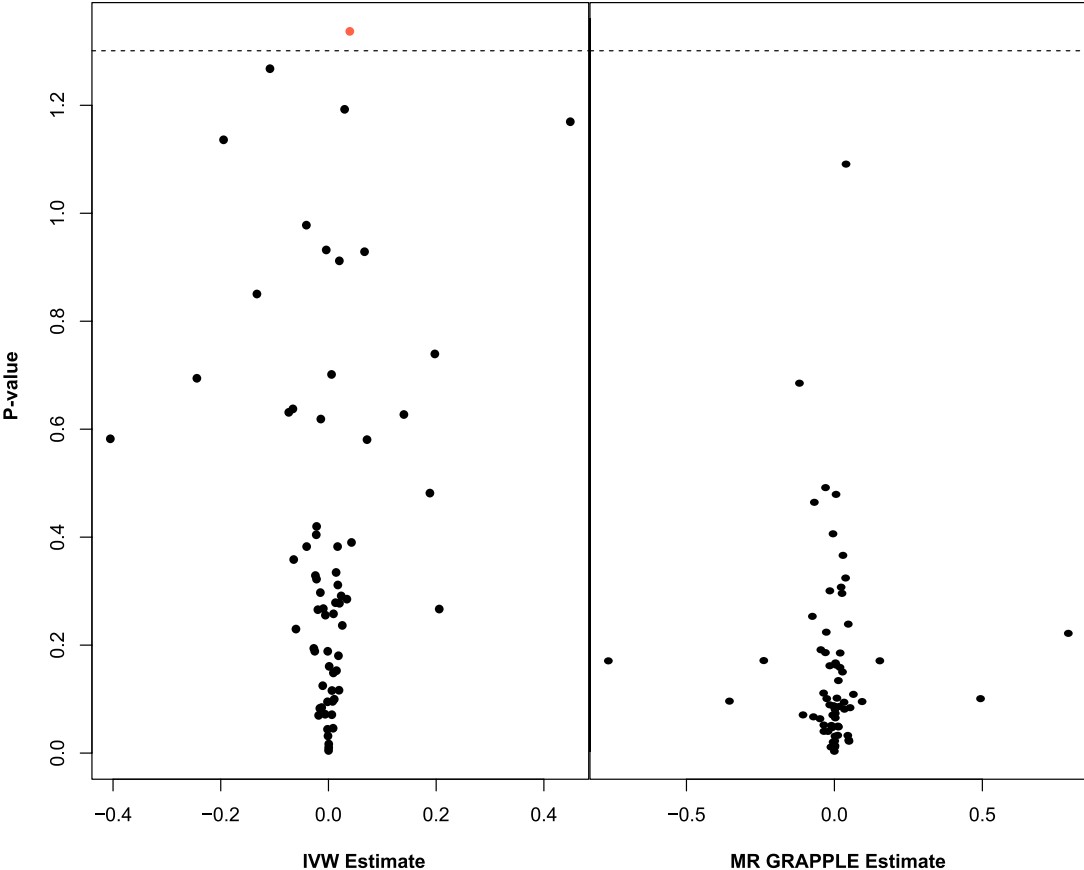

**Appendix 1—figure 4.** Multivariable Mendelian randomisation (MVMR) with IVW (left) and MVMR with GRAPPLE (*Zhao et al., 2021*) (right). Only the 66 exposures. that are significant in univariable MR (UVMR) are put forward in these models. In IVW (left), ApoB shows nominal significance. In MR GRAPPLE (right), apolipoprotein B has the lowest p-value but no trait reaches nominal significance.

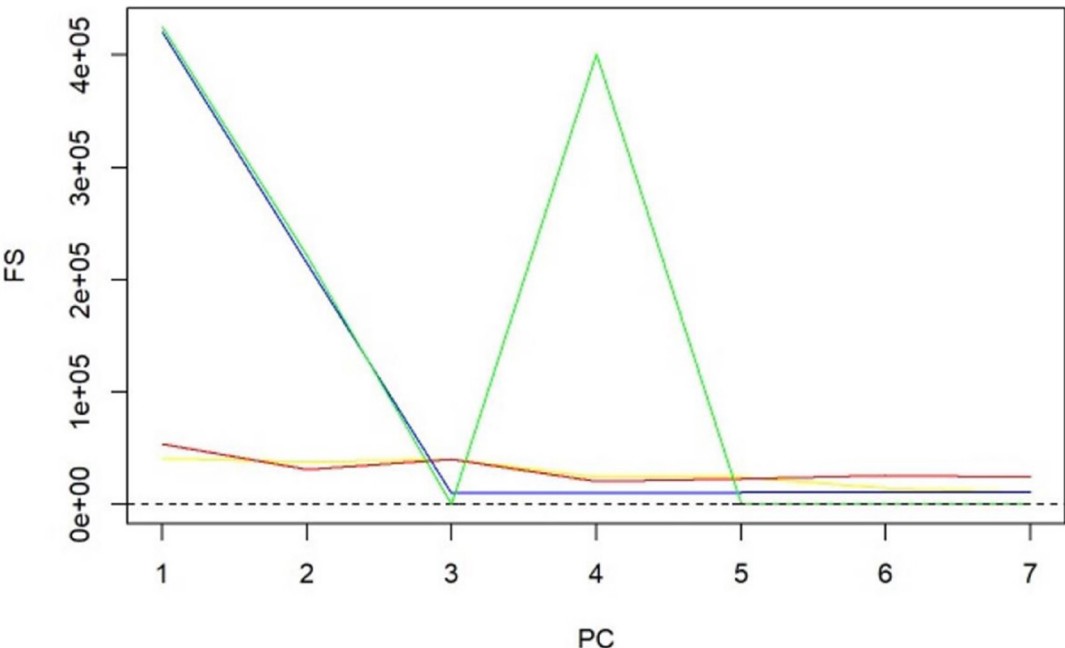

**Appendix 1—figure 5.** $F$-statistics for principal components (PCs) and sparse PCs. The formula derived in *Equation 5* is used. Black: principal component analysis (PCA) (no sparsity constraints); yellow: sparse component analysis (SCA); red: sparse PCA (Zou); blue: sparse robust PCA; green: sparse fused PCA. The dashed line represents the cutoff of 10 that is considered the minimum desired $F$-statistic for an exposure to be considered well instrumented. The green line diverges from the pattern of decreasing instrument strength but, when referring to the loadings heatmap (*Figure 2*), it can be observed that the 4th sparse PC in the fused sPCA receives negative loadings from multiple very low-density lipoprotein (VLDL)- and low-density lipoprotein (LDL)-related traits. This may in turn cause the large $F$-statistic.

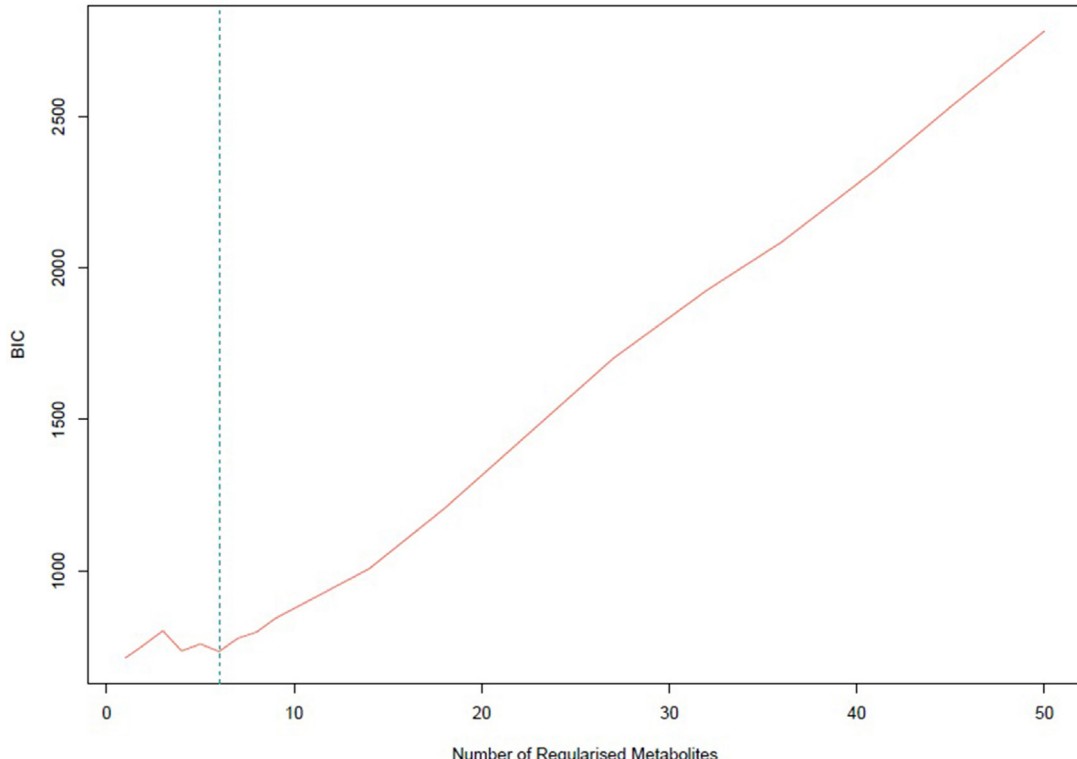

**Appendix 1—figure 6.** Bayesian information criterion (BIC) for different numbers of metabolites regularized to 0. The lowest value is achieved for one non-zero exposure per component. However, six non-zero exposures per component also achieved a similar low BIC and this was selected.

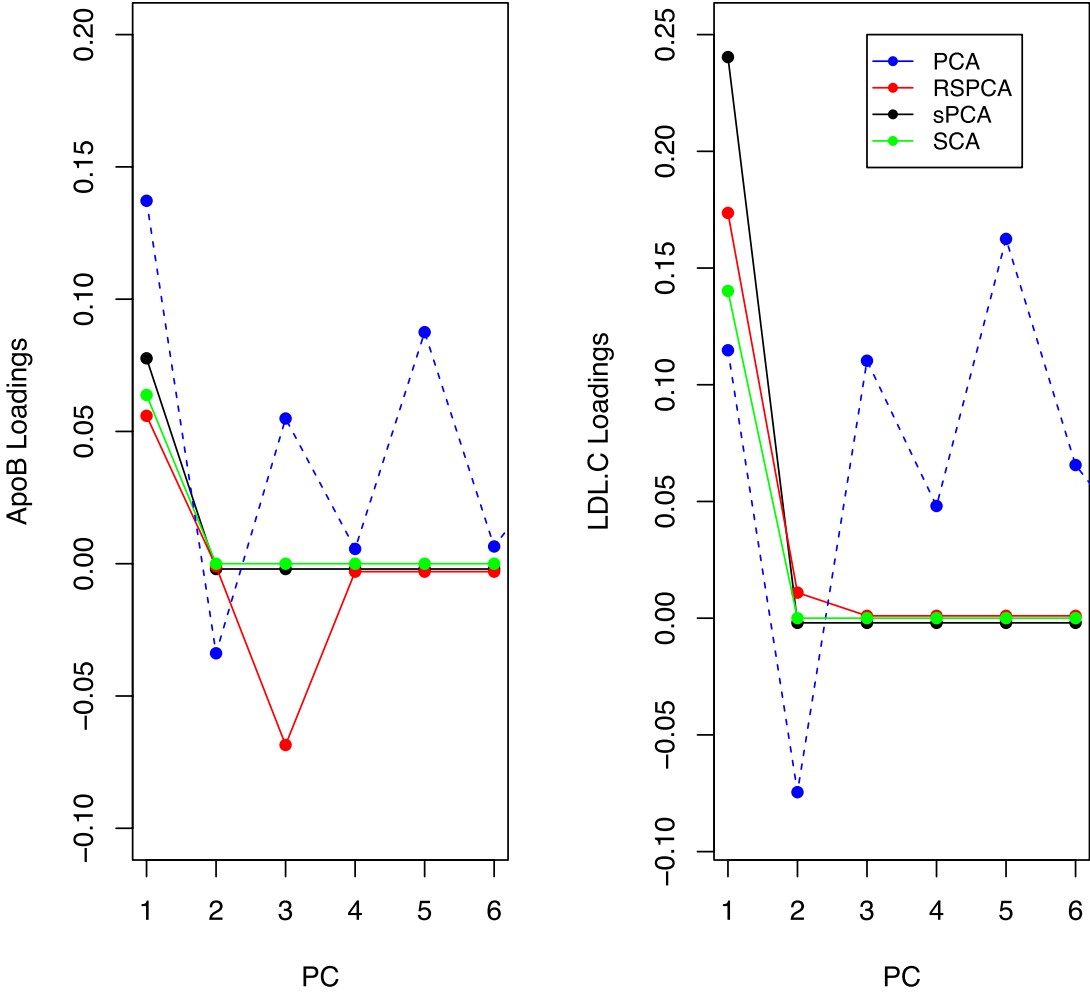

**Appendix 1—figure 7.** Trajectories for the loadings of total cholesterol in low-density lipoprotein (LDL) and ApoB in all methods. Principal component analysis (PCA) loadings imply a contribution of LDL.c and ApoB to all principal components (PCs). In the sparse methods, this is limited to one PC (two for RSPCA).

## Simulation study design

We generate a minor allele frequency $MAF \sim U(0.01, 0.5)$ and an unmeasured confounder $U \sim N(0, 5)$. Conditionally on the MAF, we generate the genes $G \sim Bin(2, MAF)$. We define $Q$ blocks and divide the $P$ genes of $G$ to $q$ blocks, each containing $P/Q$ genes. This approach aims to model the biological phenomenon of multiple traits being influenced by a common set of variants. For instance, the traits of LDL cholesterol content and LDL phospholipid content appear to be associated with the variants in a largely similar manner (**Kettunen et al., 2016**).

For each block $q = 1, 2, ..., Q$, we define a matrix $\gamma_q$ whose elements $\gamma_1 - \gamma_{P/Q}$ are non-zero and $\sim N(4, 1)$. This matrix is what will direct only certain variants to influence certain exposures, leading to multicollinearity. We then generate the $K$ exposures sequentially per block, following the parametric equation $X_q = \gamma_q G_q + U + \epsilon_X$. This way, specific genetic variants generate blocks of exposures as shown in **Figure 4** and the exposures $X_1 - X_{K/Q}$ are highly correlated. We derive the SNP-exposure associations from this dataset as $X \sim \hat{\gamma}G$. We set $K$=50–100 and $P$=100–150 and the blocks 5–8. We let these values vary across simulations in order to generate more varying values in diagnostic accuracy. For a given sample size, $s = 1000$ simulation studies were performed.

To retain the workflow of a two-sample MR, we generate a second exposure set identically as above but on an independent sample. This step is important as it guarantees the no sample overlap assumption of two-sample MR (**Burgess et al., 2016**). Based on this second $X'$ matrix, we generate the outcome $Y = \beta X' + U + \epsilon_Y$. The vector of causal estimates $\beta$ is generated based on any number of exposures being causal in the two blocks. This includes the null. We obtain the SNP-outcome

associations as $Y \sim \Gamma G$. The effect of the data generating mechanism in a single dataset is visualised in **Appendix 1—figure 8**.

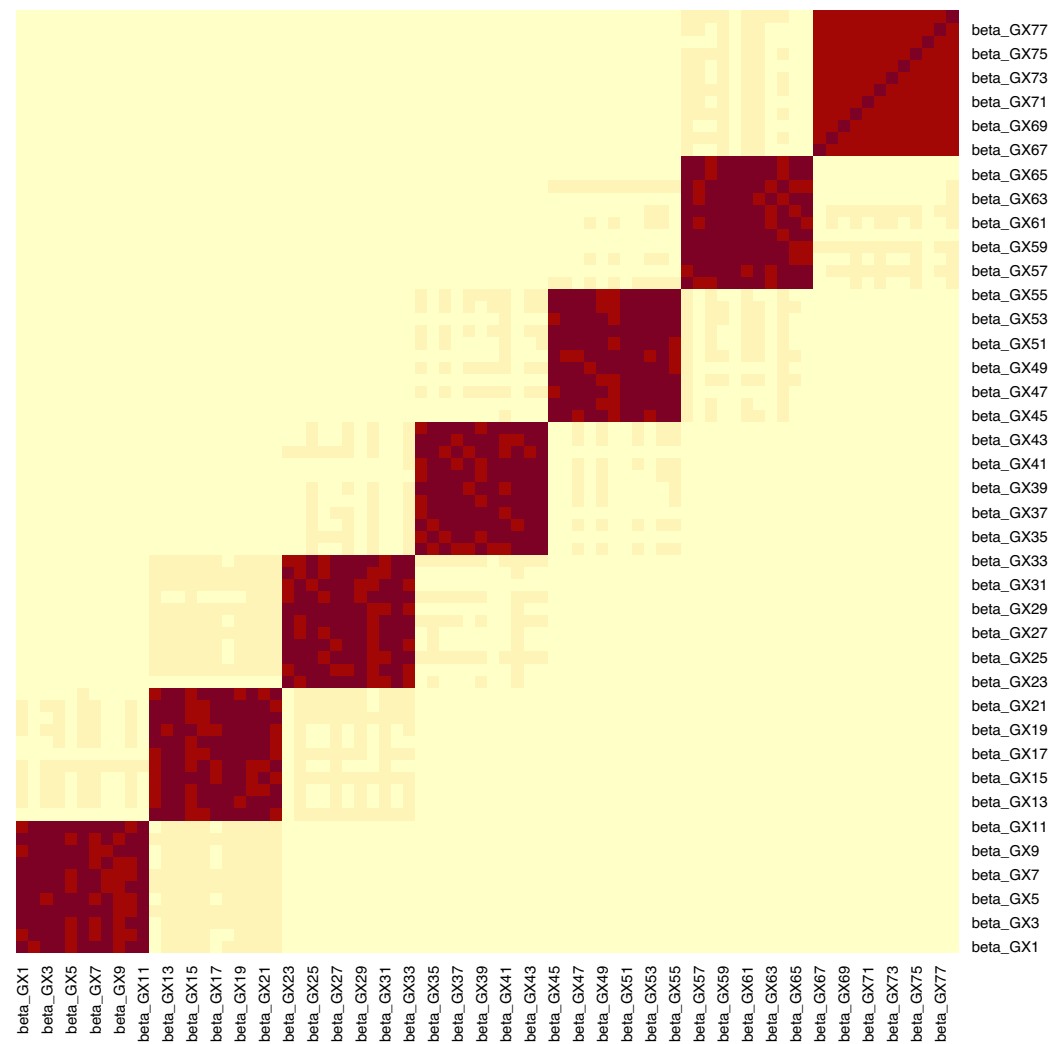

**Appendix 1—figure 8.** Example for the block correlation in $\hat{\gamma}$ ($n = 5000$, $K = 77$) induced by the data generating mechanism in **Figure 4**. In this example, the mean $F$-statistic is 231.2 and the mean **CFS** is 3.21.

The methods employed are:

- MVMR IVW with $\hat{\gamma}$ and $\hat{\Gamma}$(**Yavorska and Staley, 2020**).
- MVMR IVW with Bonferroni correction.
- PCA with the scores from $\hat{\gamma} = UDV^T$.
- sPCA as implemented in the *elasticnet* package (**Zou et al., 2006**).
- RSPCA (**Croux et al., 2013**).
- SCA (**Chen and Rohe, 2021**).

Due to computational complexity, sPCA in the *elasticnet* package (spca_en in **Figure 5**) was implemented with a simplification regarding the sparsity parameter. Specifically, it was assumed that the number of non-zero exposures per component was *P/Q*. For SCA, we use the cross-validation method in *PMA R* package (**Witten and Tibshirani, 2020**).

To generate the summary ROC curves presented in **Figure 5**, we treated simulation results as individual studies and meta-analysed them with the bivariate method of **Reitsma et al., 2005**, in the *R* package. The logit sensitivity and specificity (which are correlated) are jointly meta-analysed by modelling them as a bivariate normal distribution and employing a random-effects model for accomodating this correlation and framing it as heterogeneity.

The results for a set of simulations in $N = 3000$ individuals are presented in ***Appendix 1—figure 10***. We observe that MVMR and MVMR with Bonferroni correction estimates cluster to the bottom left of the plot, suggesting a low sensitivity. The estimates from SCA and sPCA_EN are spread in the upper left half of the plot and often achieve a sensitivity of 1.00. The PCA and RSPCA mainly provide highly sensitive estimates but perform relatively worse in specificity. The performance in increasing sample sizes is consistent with a benefit for sparse methods (***Appendix 1—figure 9***).

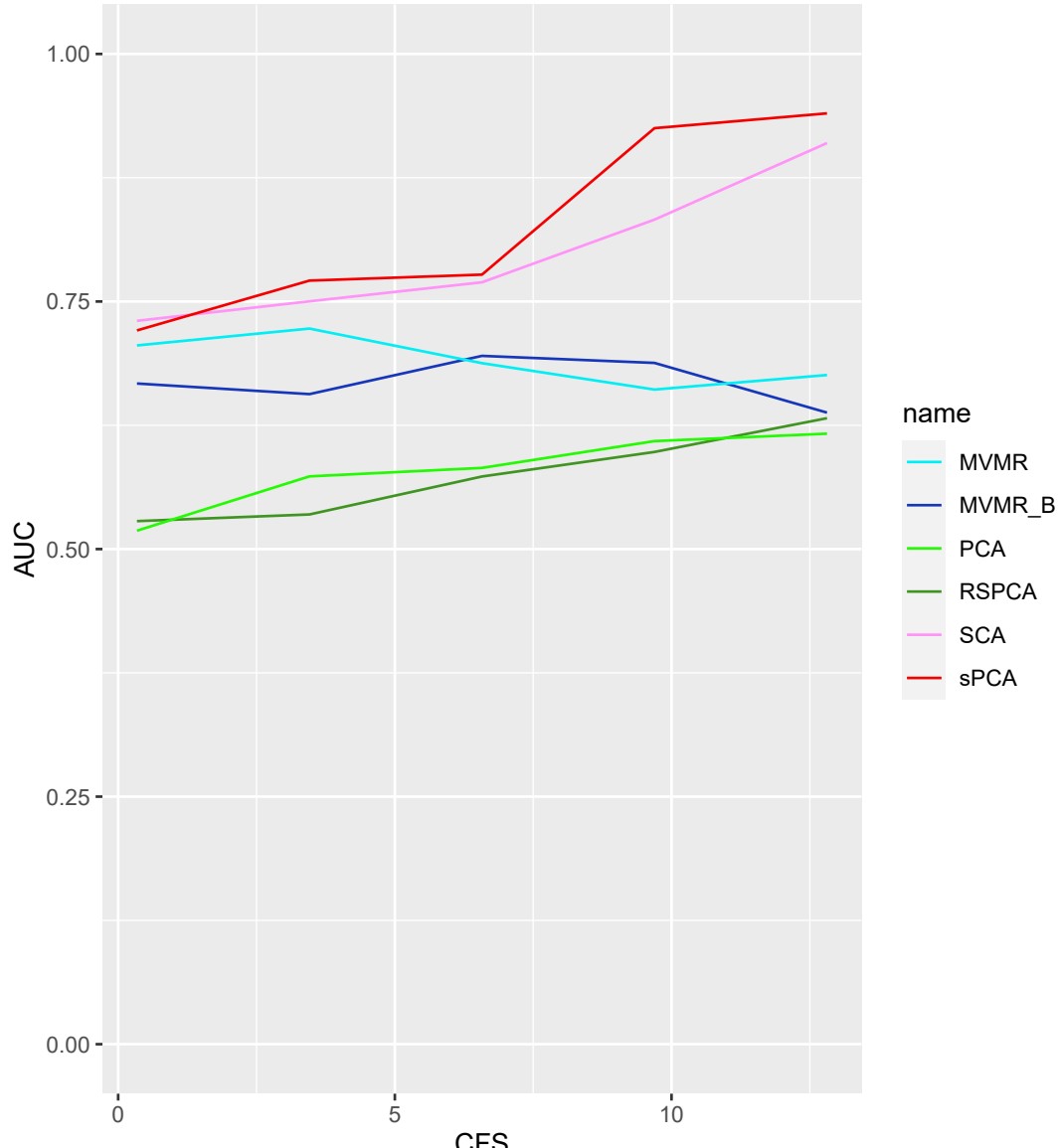

**Appendix 1—figure 9.** AUC performance of multivariable Mendelian randomisation (MVMR) and dimensionality reduction methods for increasing sample sizes. Two sparse methods (sparse component analysis [SCA], sparse principal component analysis [sPCA]) perform better compared with PCA and MVMR, with improving performance as the sample size increases. CFS: conditional *F*-statistic.

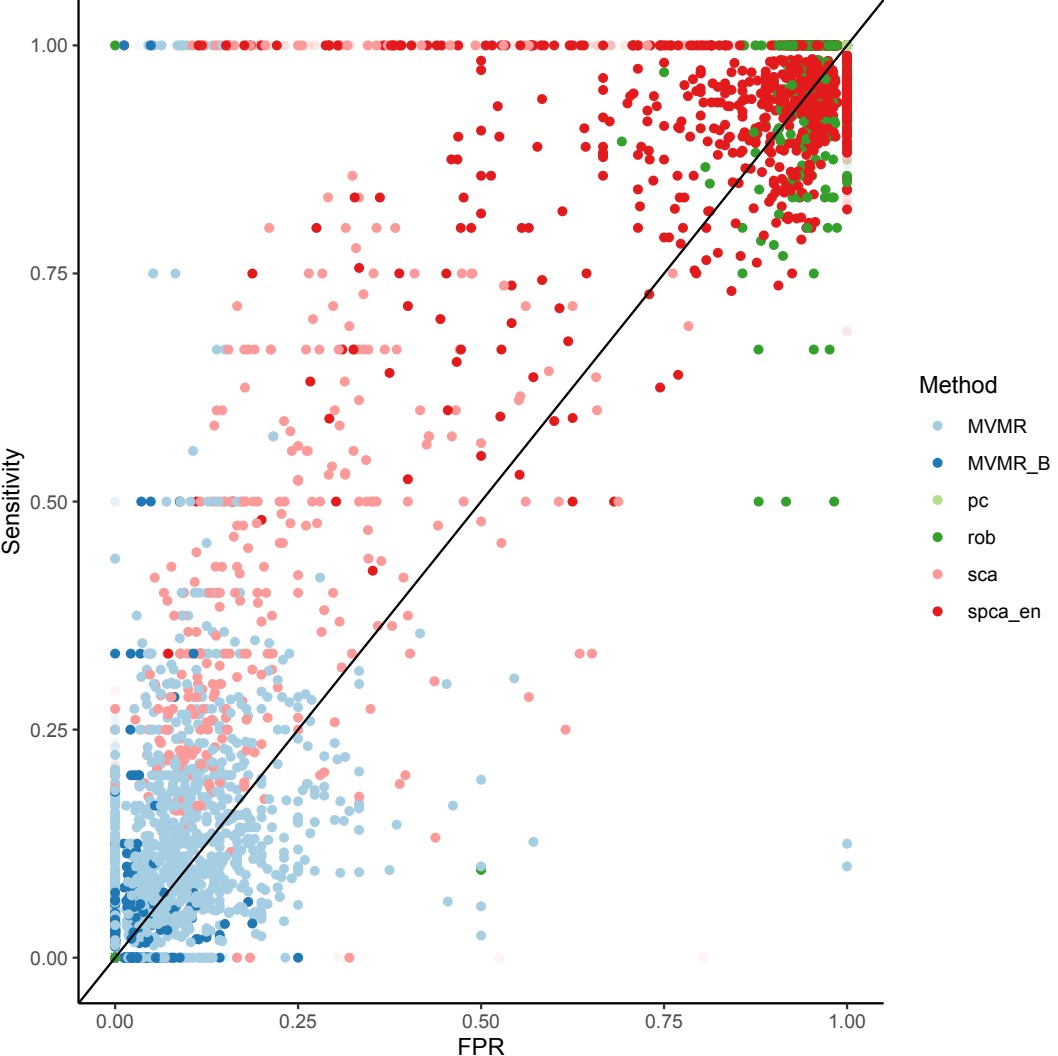

**Appendix 1—figure 10.** Individual results from $s = 1000$ simulations.

**Appendix 1—table 2.** Simulation study on only four exposures (out of the total $K = 50$) contributing to the outcome $Y$.

A drop in sensitivity and specificity is observed for sparse component analysis (SCA) and sparse principal component analysis (sPCA) compared with the simulation configuration in *Table 4*.

|  | PCA | SCA | sPCA | RSPCA | MVMR | MVMR_B |
|---|---|---|---|---|---|---|
| AUC | 0.799 | 0.714 | 0.859 | 0.492 | 0.511 | 0.675 |
| SNS | 1,0.03 | 0.75,0.25 | 1,0.17 | 0.5,0.25 | 0.25,0.25 | 0,0 |
| SPC | 0,0.2 | 0.76,0.46 | 0.66,0.18 | 0.37,0.15 | 0.94,0.07 | 1,0 |
| Youden's J | 0 | 0.428 | 0.625 | −0.029 | 0.105 | 0.032 |

## Comparison of PCA in 1SMR and 2SMR

We fix the number of blocks of exposures to $b = 2$ and the number of exposures to $K = 6$. In both 1SMR and 2SMR, the loadings matrices of $\hat{X}$ and $\hat{\gamma}_{GX}$ are of dimensionality $p \times p$ ($p$ being the number of SNPs) and can be therefore compared for similarity. We use two similarity measures: a the sum of squared differences between the one-sample and two-sample loadings matrices $S = \sum_{pj=1}^{P} \sum_{pi=1}^{P} (V_{ij,1SMR} - V_{ij,2SMR})^2$, and the $R^2$ from a linear regression of the one-sample PC

loadings on the two-sample PC loadings. $R^2$ increases and $S$ decreases with increasing sample size, as expected.

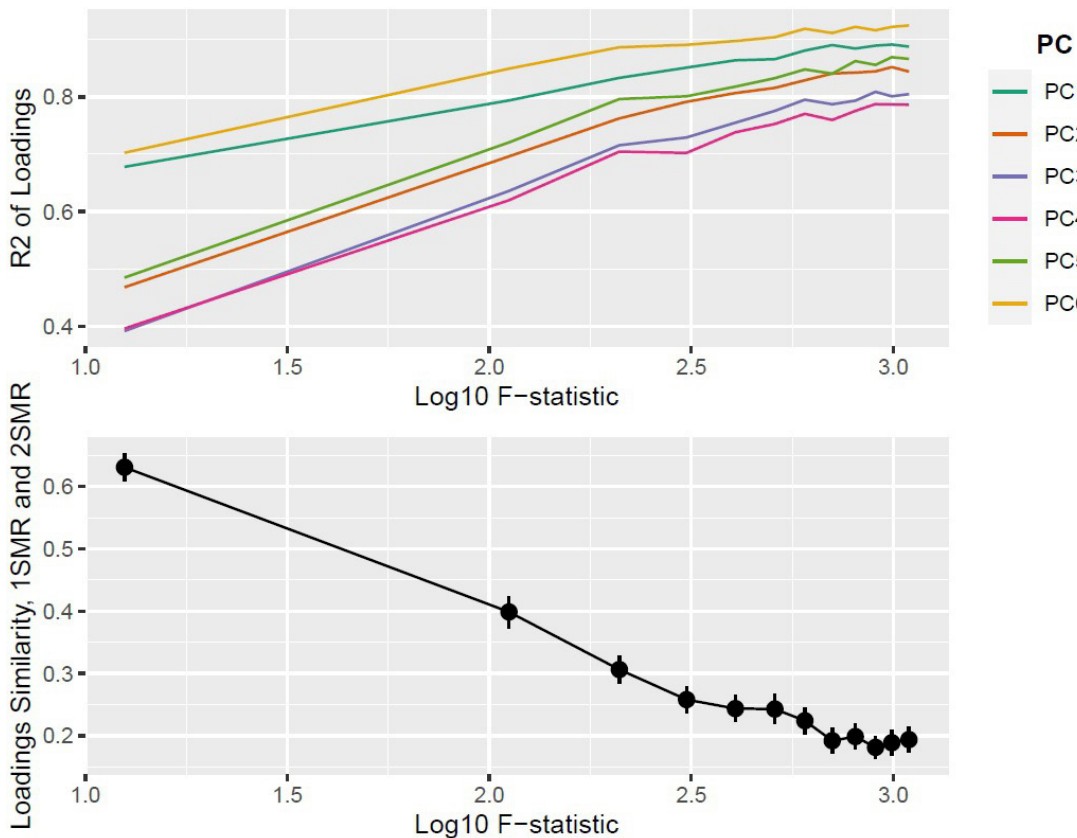

**Appendix 1—figure 11.** Top panel: $R^2$; bottom panel: similarity of loadings ($S$) between one-sample Mendelian randomisation (MR) and two-sample MR ($N_{sim} = 10,000$).

## Proportion of non-causal exposures as a predictor of performance

For a given sample size $N = 1000$, we vary the proportion of non-causal SNPs ($P_{NC}$) to examine the performance. The total number of total exposures is picked from the $K \in (100, 160)$ range, the number of SNPs from the $p \in (100, 200)$, and the number of blocks of exposures from $B \in (5, 20)$. Since the applied pipeline does not discriminate between strength of association of $X$ with $Y$ in the dimensionality reduction stage, an increase in $P_{NC}$ was expected to induce FP. Then, in the increasingly sparser subvector $\beta_b$ of the causal effects of $X_b$ on $Y$, all $X_b$ exposures would be projected to the same PC. This PC is likely to still be associated with $Y$; however, specificity may drop. This is what was observed in the simulation study, where specificity gradually drops as $P_{NC}$ increases from 20% to 80%.

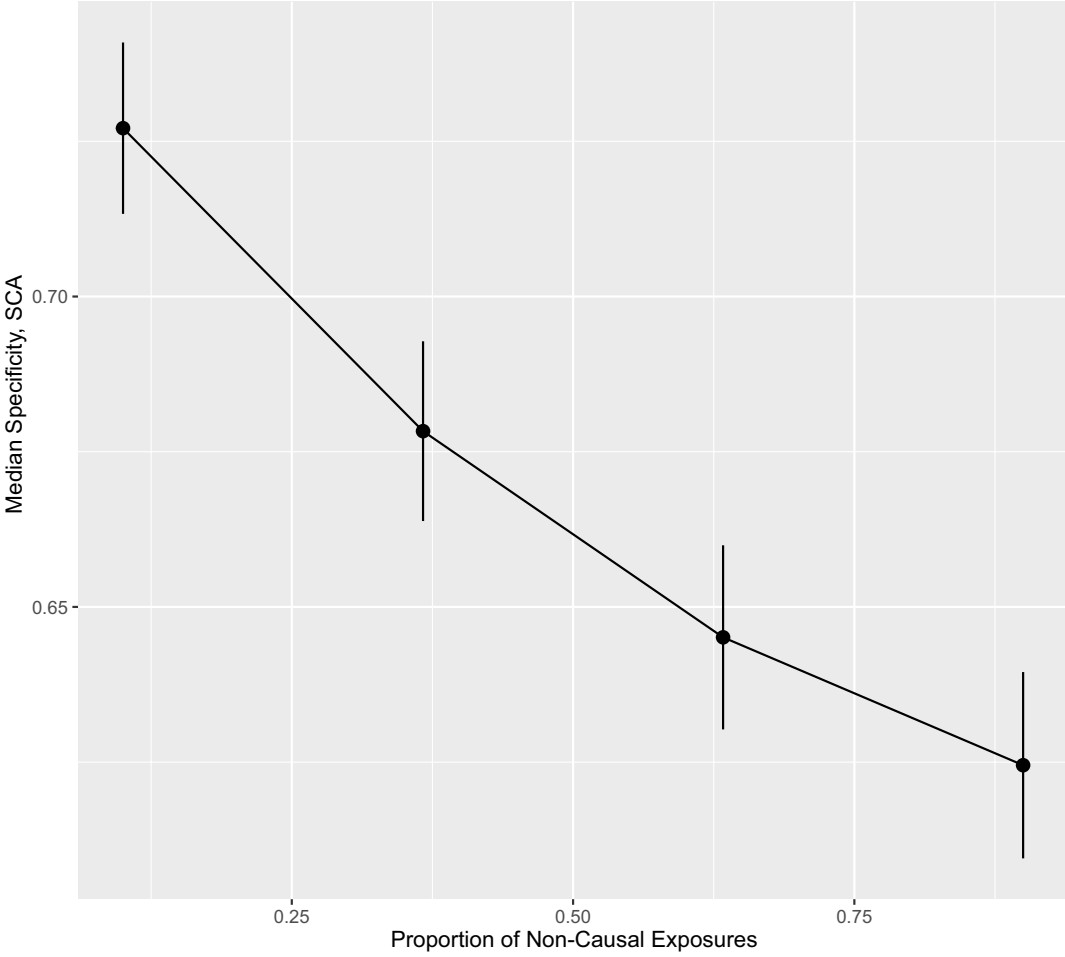

**Appendix 1—figure 12.** Specificity (ability to accurately identify true negative exposures) of sparse component analysis (SCA) as a different proportion of exposures in each block are causal for $Y$.

