## [Editor Report]

This paper investigated the identification of causal risk factors on health outcomes. It applies sparse dimension reduction methods on highly correlated traits in the Mendelian randomization framework. The implementation of this method helps to identify risk factors when given high dimensional traits data.

---

## [Decision Letter]

**Decision letter after peer review:**

Thank you for submitting your article "Sparse Dimensionality Reduction Approaches in Mendelian Randomization with highly correlated exposures" for consideration by *eLife*. Your article has been reviewed by 3 peer reviewers, and the evaluation has been overseen by a Reviewing Editor and Martin Pollak as the Senior Editor. The following individual involved in the review of your submission has agreed to reveal their identity: Stephen Burgess (Reviewer #1).

Essential revisions:

1) Clarification of the method details. All three reviewers have clarification comments.

2) Interpretation of results. See comments from reviewer 3.

3) Demonstration of benefits in real data analysis. See comments from reviewers 1 and 3.

*Reviewer #1 (Recommendations for the authors):*

I'm not the best person to judge, but I'm not sure how accessible this work would be to someone who doesn't understand multivariable Mendelian randomization. Figure 1 is a great overview, but I'd appreciate the viewpoint of someone who is less familiar with these methods.

The description of the different sparse PCA methods was a helpful summary.

I'm not sure that the foray into instrument strength is valuable – it is over 2 pages and 2 figures, and I'm not sure how much this adds – maintaining high conditional F statistics is important, but their calculation is not a major part of the analysis report. The whole section is a bit distracting and disproportionate.

All MR estimates seem to be on quite odd scales – the estimates are all close to 1 – 1.001 or 0.999. Could the PCs be standardized, so that the estimates are per 1 SD increase in the PC? This would improve communication of the results. Currently, there is no sense of scale – what is the corresponding change in lipids on a meaningful scale?

While I understand that the main thrust of the manuscript is methodological, I didn't get a strong sense of what the conclusion from the applied analysis is. If the authors could communicate a strong finding, this would help to justify the use of the method. Eg what are the estimates for the HDL clusters in the SCA method? This isn't clear.

Figure 6 would be key to this – currently, it's not clear what the various PCs represent. Maybe a radar chart would help? Also, the caption for Figure 6 is incorrect – the caption from Figure 8 has inadvertently been duplicated here.

While I wouldn't want to insist that the authors change their paper, the obvious application of this work appears to be imaging studies – this is currently under-explored.

Figure 5 is a really helpful summary of the performance of the different methods in this investigation.

*Reviewer #2 (Recommendations for the authors):*

This is a nice paper, with convincing and very useful results. I have mainly some clarification questions and suggestions.

1. The main idea is to treat the matrix of SNP-exposures associations γ^ as explanatory variables in the regression Γ^=γ^β+e, although this is clearly not a standard regression model, just that a WLS or GLS estimator for *β* is equivalent to the 2sls estimator on single-sample data. I think it would therefore be instructive to link to the construction of PCs in a single-sample individual-level data sets. There the model is y=Xβ+uX=Gγ+V and one could do a PCA on **X**. Alternatively, as the 2sls estimator is least-squares in y=X^β+u~ where X^=Gγ~, one could perhaps consider PCA on X^. As X^TX^=γ^TGTGγ^ this is getting close to the PCA on γ^ if GTG is normalized, for example GTG/n = **I**_p_. If it is not normalized, then it is hard to make a connection. It may therefore be worthwhile considering PCA on the weighted γ^ as per the (extended) IVW estimator in (1) on page 4. With such a link in place, it seems easier to motivate the proposed method.2. Are the covariances σlkj reported in the GWAS? I assume the estimators are simply IVW throughout, as implied at the top of page 5?3. It is unclear to me why ∆ is estimated by OLS in (2).

4. Notation. It would be easier to follow the notation if matrices are upper case and vectors lower case. That would change for example the current vector **γ** to a lower case letter and the current matrix **γ** to an upper case letter.

5. It is not straightforward to follow the steps for Sparse PCA on page 56, as not all symbols are clear. It is not clear in 1. what "Setting a fixed matrix," means and what the α_j_ are. Also, define **Xi**. In 2 you simply do an svd? Should there be "hats" on **γ**? UDV^T^ in step 2. are not the same values as in the svd for γ^.

*Reviewer #3 (Recommendations for the authors):*

Here are some detailed comments.

1. Usually a good idea to number figures in the order they appear in the text (Figure 12 out of order).

2. UVMR is referred to as UMVR three times and as UMVMR once

3. At the beginning of the Methods section, it would be good to specify the dimension of \γ. It is also important to specify how the variants used are selected.

4. Figure 1 is a bit chaotic and inconsistent with the text. Figure 1 top, the graph indicates that there are 118 exposures. I assume this was chosen because it is the number of lipid exposures in the data application but it doesn't look like the lipid data is used in this figure. Figure 1 top right, the dimension of the matrix is given as L x 118. The variable L does not appear elsewhere in the paper. What is printed appears to be a 60x60 square matrix. Is this matrix meant to represent γ⊤γ? The statement "MR estimates affected by multi-colinearity" printed next to this matrix doesn't appear to relate to the figure because the figure doesn't indicate anything about the MR estimates.5. Figure 5. It looks like none of the matrices shown have all 118 rows. The sparse PCA matrix has many fewer exposures than the other methods. It would be good to clarify this in the caption.

6. Figure 6 is labeled as "Extrapolated ROX curves" but appears to be a comparison of UVMR and MVMR results. For the top right plot, I would have expected UVMR and MVMR results to be identical since the PCs are exactly orthogonal. Why is this not the case?

7. Page 12 describes Figure 5 as "The loadings for all methods (Figures 5a-e) are presented for LDL.c, a previously prioritised trait." I am confused about what "for LDL.c" means here. It appears that LDL.c is one of the traits in Figure 5 but these appear to be loadings across some large subset of traits?

[Editors' note: further revisions were suggested prior to acceptance, as described below.]

Thank you for resubmitting your work entitled "Sparse Dimensionality Reduction Approaches in Mendelian Randomization with highly correlated exposures" for further consideration by *eLife*. Your revised article has been evaluated by Martin Pollak (Senior Editor) and a Reviewing Editor.

The manuscript has been improved and all reviewers agree this study will be useful for the field. However, there are remaining issues and we would like to invite you to further revise this manuscript.

Revision outline:

1. Adding a deeper more nuanced discussion of what is meant by causality and how to interpret the components. The major advancement of this study comes from method development and it is important to not overstate the significance of results. See comment 1 from reviewer 3 and comment 4 from reviewer 1.

2. The choice of method. Discussing scenarios that will work or fail for the method will be helpful for the readers to better utilize this tool. See comment 4 from reviewer 1 and comment 2 from reviewer 3.

3. Adding details of simulation. Other clarification issues raised by reviewers.

*Reviewer #1 (Recommendations for the authors):*

1. While I appreciate the simulation study, and believe this is worthwhile, it strikes me that the primary reason for choosing one or other of these sparse methods is not performance in a simulation study, but rather how the methods define the PCs, and how the analyst could interpret the results. As there's no point running an analysis if you can't interpret the results. This will depend on the specific characteristics of the investigation, so I am not expecting a single method recommendation. But I'd appreciate a bit more acknowledgement of this in the choice of methods – what characteristics to these methods tend to result in? Interesting (and reassuring) to see that clusters from the SCA and sPCA methods look similar in Figure 2 (although then odd that in Table 3, the significance differs for the VLDL and LDL components…).

Overall, I appreciate the authors' manuscript in that it proposes sparse dimension reduction as a method and explores the properties of MVMR estimates from this approach. My sense is that it is difficult to say anything more authoritative/general than "hey, look – we tried this and it sort of worked!", but the authors do this in a way that is mostly clear (see comments above) and should be helpful to future researchers.

*Reviewer #2 (Recommendations for the authors):*

I am happy with this revision. The authors have responded well to my previous comments and suggestions, or clarified their preferred choices.

*Reviewer #3 (Recommendations for the authors):*

The resubmission has corrected some of the issues I raised previously. However, I have some remaining concerns, primarily about the interpretation of the proposed causal effects and the simulation results.

1. My largest issue is still in the causal interpretation of genetically determined PCs are sparse components. To me, I don't think there is a way to turn the components into well-defined exposures that admit a counterfactual or interventional definition of a causal effect. This approach seems closer to "causality adjacent" methods like TWAS. There was a paper a few years ago using sparse CCA in combination with TWAS (https://journals.plos.org/plosgenetics/article?id=10.1371/journal.pgen.1008973) and this seems similar to that to me. I think one option for this paper would be to add a deeper discussion of what it means to test for a causal effect of a component that may or may not represent a latent biological variable. On one end of interpretation you have "association with genetically predicted exposure component" and then I think you could add as a bonus that if you have successfully identified a biological component, this could be interpretable as a causal effect.

2. Using the components introduces an issue of horizontal pleiotropy if a single variant is allowed to affect multiple components. This suggests that the component-based MVMR results would be more appropriate than component-based UVMR results, though for PCA this is not an issue due to orthogonality and approximate equivalence of UVMR and MVMR.

Some questions about the simulations:

3. In Figure 4, it seems like the MVMR plot should have 6 lines and not two. How were the 6 traits combined into two lines here? It seems unfair to average the rejection rate for the X_1-3 into the black line since, if MVMR performed perfectly, it would never reject X_3 and always reject X_1 and X_2.

4. Why is there an interval around only the black line in the MVMR plot and not the other plots?

5. The sentence in the caption presumably describing the color scheme is cutoff. "The results are split according to group n black…"

6. Is the appropriate number of components known a-priori in the simulations?

7. The description of Figure 4 suggests that PCA and SCA are out-performing MVMR. However, PCA has exactly the same rejection rate for the causal and non-causal block of traits which seems like bad performance to me. SCA has a Type 1 error rate between 20 and 30% at a nominal level of 0.05 which also seems quite bad to me. I would not view improving power at the cost of uncontrolled type 1 error as a good outcome.

8. The description of the Figure 5 at the top of page 9 does not align with the image in Figure 5. "Both standard and Bonferroni-corrected MVMR performed poorly in terms of AUC (AUC 0.660 and 0.885 respectively), due to poor sensitivity. While there was a modest improvement with PCA (AUC 0.755), it appears that PCA and RSPCA do not accurately identify negative results (median specificity 0 and 0.28 respectively)." Reading these numbers, it sounds like MVMR is the worst followed by PCA, followed by MVMR_B. However, in the figure, PCA and RSPCA both have lower AUC than MVMR and MVMR_B. These numbers are also inconsistent with Table 4.

9. I think I may be a little confused about how the ROC curves are constructed.

9a. In my experience, an ROC curve is usually constructed by varying a threshold (e.g. α). However, it seems here like α was fixed and we are averaging over the single point obtained for each simulation setting according to the Reitsma et al. method. This method makes sense when the data are real observations, however, in the case of simulations, why not use the traditional method of varying to threshold and then averaging the full curves across simulations?

9b. I think this method of displaying results also obscures the issue of calibration. If there is a nominal significance threshold, we would like to see that type 1 error is controlled below that threshold.

9c. Visually, Figure 15 looks inconsistent with Figure 5 to me. There are no blue points in the upper right of Figure 15 -- does it make sense to extrapolate this part of the curve? Figure 5 suggests that the red methods at least have the potential to do very well in differentiating true and false positives but this doesn't seem born out by Figure 15 where most of the red points cluster in the upper right.

10. On the selection of instruments -- is it correct that you are assuming that all variants affecting any of the component metabolites also affect LDL/HDL or Triglycerdies as measured in the GLGC study? Does this assumption limit your ability to identify components that separate LDL or HDL sub-components? For example, variants that affect only a small number of components will be less likely to be selected using the GLGC reference.

---

## [Author Response]

Reviewer #1 (Recommendations for the authors):I'm not the best person to judge, but I'm not sure how accessible this work would be to someone who doesn't understand multivariable Mendelian randomization. Figure 1 is a great overview, but I'd appreciate the viewpoint of someone who is less familiar with these methods.

We have expanded the Introduction (first and second paragraphs), stressing the link of MR with RCTs that are more widely known. The goal of confounding adjustment is very similar in the two approaches and we believe this can motivate the presented methods for a wider audience.

The description of the different sparse PCA methods was a helpful summary.I'm not sure that the foray into instrument strength is valuable – it is over 2 pages and 2 figures, and I'm not sure how much this adds – maintaining high conditional F statistics is important, but their calculation is not a major part of the analysis report. The whole section is a bit distracting and disproportionate.All MR estimates seem to be on quite odd scales – the estimates are all close to 1 – 1.001 or 0.999. Could the PCs be standardized, so that the estimates are per 1 SD increase in the PC? This would improve communication of the results. Currently, there is no sense of scale – what is the corresponding change in lipids on a meaningful scale?

Thank you for your suggestion, indeed the OR estimates are on a narrow scale close to 1. We relate this to the lack of definition of the estimated causal effects. For this reason, we limit the scope of the proposed method to hypothesis testing as opposed to estimation. It is encouraging that, in the positive control example of how lipids associate with CHD, we observe that, when positive loadings are used to transform the exposures, the sign is biologically meaningful (LDL and VLDL increase risk, HDL appears to be protective). We acknowledge this and further stress its implications in the Discussion/Limitations section.

While I understand that the main thrust of the manuscript is methodological, I didn't get a strong sense of what the conclusion from the applied analysis is. If the authors could communicate a strong finding, this would help to justify the use of the method. Eg what are the estimates for the HDL clusters in the SCA method? This isn't clear.

Thank you for your suggestion, indeed there is an interesting point not originally highlighted regarding the HDL cluster. In the MVMR with those exposures that were significant in UVMR, only ApoB is positively associated with CHD. No exposures were protective. However, in SCA, there is a grouping of HDL traits in two independent PCs, corresponding to small/medium and larger HDL traits. The former PC appears to be protective for CHD. This finding is supported from other observational studies (smaller HDL traits are associated with favourable cardiovascular risk profiles) that we now report in the Discussion. Thus, if the analysis was restricted to MVMR, these findings would be masked due to imprecision (additions in Discussion/First Paragraph, updated Table 4).

Figure 6 would be key to this – currently, it's not clear what the various PCs represent. Maybe a radar chart would help? Also, the caption for Figure 6 is incorrect – the caption from Figure 8 has inadvertently been duplicated here.

Thank you for your observation, we have fixed the caption. The figure has also slightly changed following the comments of Reviewer 3 on an update in the exposures that are eligible with the SNPs that we use for the instrument. What Figure 6 reports is the comparison of the univariable and multivariable estimates of the causal effect of each PC on CHD. We would expect PCA to provide orthogonal components and therefore to have identical estimates in UVMR and MVMR. In the sparse methods, where strict orthogonality is traded with interpretability of the PCs, we expected some deviation and we wanted to visually communicate this. We also included the UVMR and MVMR estimates for SFPCA which were not in the original draft.

While I wouldn't want to insist that the authors change their paper, the obvious application of this work appears to be imaging studies – this is currently under-explored.

Thank you for your observation, we agree that imaging data would be a very good candidate for this approach. We highlight this in the Discussion.

Reviewer #2 (Recommendations for the authors):This is a nice paper, with convincing and very useful results. I have mainly some clarification questions and suggestions.1. The main idea is to treat the matrix of SNP-exposures associations γ^ as explanatory variables in the regression Γ^=γ^β+e, although this is clearly not a standard regression model, just that a WLS or GLS estimator for β is equivalent to the 2sls estimator on single-sample data. I think it would therefore be instructive to link to the construction of PCs in a single-sample individual-level data sets. There the model is y=Xβ+uX=Gγ+V and one could do a PCA on **X**. Alternatively, as the 2sls estimator is least-squares in y=X^β+u~ where X^=Gγ~, one could perhaps consider PCA on X^. As X^TX^=γ^TGTGγ^ this is getting close to the PCA on γ^ if GTG is normalized, for example GTG/n = **I**_p_. If it is not normalized, then it is hard to make a connection. It may therefore be worthwhile considering PCA on the weighted γ^ as per the (extended) IVW estimator in (1) on page 4. With such a link in place, it seems easier to motivate the proposed method.

Thank you for your suggestion. We now include a simulation study on one-sample data where we perform PCA (*PCA*_1*SMR*_) and one sparse implementation (SCA) on X^. To accomodate the larger computational time (matrix of *n* × *K* instead of *p* × *n*), we reduced sample size and accordingly increased the *γ* parameter, keeping the *F*−statistics and conditional *F*−statistics in the same range. We compare a two-stage least squares MVMR estimator for *K* exposures with the following procedure. We obtain PC scores from *X*ˆ = *UDV ^T^* and then regress *UD* on *Y*. Figure XYZ shows a similar performance with individual-level data as the one reported in the original draft. Both MVMR and MVMR with a multiplicity correction suffer from imprecise results (low sensitivity in Figure). SCA performs better in identifying true causal exposures, whereas PCA provides a lot of false positive results. Therefore, we conclude that such an approach with two-stage least squares is feasible and performs better than PCA in the simulation study.

We then examine how similar *PCA*_1*SMR*_ and *PCA*_2*SMR*_ are. As suggested, the aim is to establish how much information is lost by opting for a PCA/SCA on summary data only. If the exposures are grouped together in *PCA*_1*SMR*_ and *PCA*_2*SMR*_ in a similar fashion, then this would indicate a similar performance in individual level data and summary statistics. To quantify this, we keep the simulation setup described in the *Simulation Study* Section and we apply the two methods.

As you suggested, the loadings matrices for both is of dimensionality *K* × *K*. We then use two measures to quantify the similarity for increasing sample sizes: a similarity measure to compare the loadings matricesSload=∑k1=1K∑k2=1K(Vk1k2,1SMR−Vk1k22SMR)2,

where *V*_1*SMR*_ are the loadings assigned in one-sample MR and *V*_2*SMR*_ those of two-sample MR, and (b) the *R*^2^ of the regression of Vk1k2,1SMR on Vk1k22SMR (R2=1−ϵVVTϵVVV1SMRTV1SMR, where *ϵ_V V_* are the residuals of the regression of *V*_1*SMR*_ on *V*_2*SMR*_). We vary the sample size and track the performance of PCA and SCA.

The results are presented in Figure 1. For SCA, *PCA*_1*SMR*_ and *PCA*_1*SMR*_ load the matrix in a more similar manner as sample size increases, however not converging to zero. A similar picture is drawn by the *R*^2^, with all PCs showing a closer alignment of 1SMR and 2SMR as sample size increases. A closer examination of the results in individual simulations suggests that the blocks of exposures are correctly projected to PCs (exposures 1 to 3 in one PC and 4 to 6 in another, Figure), more so in larger sample sizes We note that the loadings for each block are not identical in 1SMR and 2SMR.

As suggested, in the applied analysis we have replaced the PCA on *γ* approach with the one proposed. We thus standardise γ as.γGXSEγGX There are no differences in precision of the causal estimates.

2. Are the covariances σlkj reported in the GWAS? I assume the estimators are simply IVW throughout, as implied at the top of page 5?

We have used the intercept of the LD score regression as a proxy for the phenotypic correlations *σ_lkj_*.

We clarify that, if individual-level data is not available, this is a sensible approach. Yes, IVW is used throughout the analyses.

**Author response image 1. sa2fig1:** Top Panel: *R*^2^; Bottom Panel: Similarity of loadings (*S_load_*) between one-sample MR and two-sample MR (*N_sim_* = 10*,*000).

3. It is unclear to me why ∆ is estimated by OLS in (2).

Thank you for your comment, δ^ is estimated by regressing an exposure *X_k_* on the other exposures *X*_−*k*_ by OLS. If an exposure *k* can be accurately predicted conditionally on other exposures, then there won’t be sufficient variation in *Q_Xk_* (low conditional instrument strength). We have clarified this in the manuscript.

4. Notation. It would be easier to follow the notation if matrices are upper case and vectors lower case. That would change for example the current vector γ to a lower case letter and the current matrix γ to an upper case letter.

Thank you for your comment, we prefer to refer to the SNP-*X* associations as *γ* and to the SNP-*Y* associations as Γ as this is common practice.

5. It is not straightforward to follow the steps for Sparse PCA on page 56, as not all symbols are clear. It is not clear in 1. what "Setting a fixed matrix," means and what the α_j_ are. Also, define Xi. In 2 you simply do an svd? Should there be "hats" on γ? UDV^T^ in step 2. are not the same values as in the svd for γ^.

Thank you for your comment, we have updated the explanation on sparse PCA in page 5. We are rephrasing the page 272 of the Zou et al., 2006 paper that introduces the ideas of framing PCA as a ridge regression problem (*L*2 norm is similar to variance maximisation) and then additionally using an *L*1 norm to induce sparsity.

Reviewer #3 (Recommendations for the authors):Here are some detailed comments.1. Usually a good idea to number figures in the order they appear in the text (Figure 12 out of order).

Thank you for spotting this, we have reordered the Figures.

2. UVMR is referred to as UMVR three times and as UMVMR once

Thank you for spotting this, we have corrected this.

3. At the beginning of the Methods section, it would be good to specify the dimension of γ. It is also important to specify how the variants used are selected.

Thank you for your suggestion, we have added the dimensions of γ (148x118). The instrument selection procedure is described in the first paragraph of the Methods/Data section.

4. Figure 1 is a bit chaotic and inconsistent with the text. Figure 1 top, the graph indicates that there are 118 exposures. I assume this was chosen because it is the number of lipid exposures in the data application but it doesn't look like the lipid data is used in this figure. Figure 1 top right, the dimension of the matrix is given as L x 118. The variable L does not appear elsewhere in the paper. What is printed appears to be a 60x60 square matrix. Is this matrix meant to represent \γ^\top \γ? The statement "MR estimates affected by multi-colinearity" printed next to this matrix doesn't appear to relate to the figure because the figure doesn't indicate anything about the MR estimates.

Thank you for looking into this Figure, we indeed do not use the lipid data in this. Instead, we use simulated data. To improve the Figure, we have switched to the updated exposures γ*^T^γ* matrix (147 × 99). We changed the notation to *K*, which is how we refer to the number of exposures throughout the text. We have moved the ’MR Estimates affected by multi-collinearity’ sentence to the legend.

5. Figure 5. It looks like none of the matrices shown have all 118 rows. The sparse PCA matrix has many fewer exposures than the other methods. It would be good to clarify this in the caption.

Thank you for your suggestion, we now clarify this in the caption.

6. Figure 6 is labeled as "Extrapolated ROX curves" but appears to be a comparison of UVMR and MVMR results. For the top right plot, I would have expected UVMR and MVMR results to be identical since the PCs are exactly orthogonal. Why is this not the case?

Thank you, we have corrected the caption. This is correct, we are comparing UVMR and MVMR estimates for the PCA methods. We agree and also hypothesised that the estimates would be identical for PCA where the PCs are strictly orthogonal and similar in the sparse implementations. In the updated analysis where aminoacids are excluded and six (instead of seven) PCs are constructed, this is also observed. The agreement of the causal estimates from PCA is the highest (*R*^2^ = 0*.*93). The sparse methods provide estimates that are not as concordant as in PCA but are still highly correlated. Regarding the poor performance of robust sparse PCA on this assessment, we have increased the number of sparsity parameters per component and this appeared to improve (*R*^2^ = 0*.*79).

7. Page 12 describes Figure 5 as "The loadings for all methods (Figures 5a-e) are presented for LDL.c, a previously prioritised trait." I am confused about what "for LDL.c" means here. It appears that LDL.c is one of the traits in Figure 5 but these appear to be loadings across some large subset of traits?

Thank you, we were presenting separately the loadings that corresponded to LDL.c and urea across all methods in Figure 12. Figure 5 was a reference to the methods but this referencing was confusing so we dropped it. In the updated Figure, we present visually how the different methods project LDL.c and ApoB to different components; PCA loadings suggest that LDL.c contributes to some degree to all PCs whereas the sparse methods reduce this to a single PC.

[Editors' note: further revisions were suggested prior to acceptance, as described below.]

The manuscript has been improved and all reviewers agree this study will be useful for the field. However, there are remaining issues and we would like to invite you to further revise this manuscript.

Thank you for your positive assessment of our manuscript. We have updated the manuscript according to the reviewers’ comments and have included our responses point-by-point in the present document.

We also like to point out a change in the senior authorship, with Prof. Jack Bowden and Dr. Verena Zuber now both being joint senior authors, as Prof. Jack Bowden contributed to the manuscript to a degree that qualifies him for this. All authors agreed to this author order.

Revision outline:1. Adding a deeper more nuanced discussion of what is meant by causality and how to interpret the components. The major advancement of this study comes from method development and it is important to not overstate the significance of results. See comment 1 from reviewer 3 and comment 4 from reviewer 1.

Thank you for your suggestion. We have updated the interpretation of the findings in line with these two comments. We stress more on how the methodological improvements are the major focus.

2. The choice of method. Discussing scenarios that will work or fail for the method will be helpful for the readers to better utilize this tool. See comment 4 from reviewer 1 and comment 2 from reviewer 3.

Thank you for your point. We have expanded the Discussion section to describe the types of datasets that the proposed methods are applicable for. We believe this makes things clearer for practitioners.

2. Adding details of simulation. Other clarification issues raised by reviewers.

Thank you for your suggestion. We have substantially rephrased and reorganised the simulation section, with changes highlighted in blue. We have added the required details and split the simulation results in subsections to improve clarity.

Reviewer #1 (Recommendations for the authors):1. While I appreciate the simulation study, and believe this is worthwhile, it strikes me that the primary reason for choosing one or other of these sparse methods is not performance in a simulation study, but rather how the methods define the PCs, and how the analyst could interpret the results. As there's no point running an analysis if you can't interpret the results. This will depend on the specific characteristics of the investigation, so I am not expecting a single method recommendation. But I'd appreciate a bit more acknowledgement of this in the choice of methods – what characteristics to these methods tend to result in? Interesting (and reassuring) to see that clusters from the SCA and sPCA methods look similar in Figure 2 (although then odd that in Table 3, the significance differs for the VLDL and LDL components…).

Thank you for your comment. We agree that, given the study design, final recommendations on using only one method could be missing out on subtle differences; for instance, datasets with outliers might benefit more from robust approaches but this approach didn’t seem to outperform others in our simulation study nor in the applied example. We show a substantial improvement in performance for the particular realistic simulation design that we describe, and the performance measure is based on hypothesis testing rather than how the methods define the PCs. The simulation study is now reorganised with three distinct sections and we believe it is clearer how we do measure the results of how the PCs affect the outcome (power and Type I Error).

Regarding the Table 3 results, we agree that the sPCA results are not perfectly accordant with the expectation of a LDL/VLDL positive association with CHD. We used this example as a positive control outcome and we therefore interpret this as a suboptimal performance of sPCA in this particular context. It is of course possible to imagine different scenarios regarding this difference with the simulations where sPCA performed well, the main of whom would be some form of pleiotropy. We discuss this possibility in the discussion.

Overall, I appreciate the authors' manuscript in that it proposes sparse dimension reduction as a method and explores the properties of MVMR estimates from this approach. My sense is that it is difficult to say anything more authoritative/general than "hey, look – we tried this and it sort of worked!", but the authors do this in a way that is mostly clear (see comments above) and should be helpful to future researchers.

Thank you for your positive assessment of our manuscript.

Reviewer #3 (Recommendations for the authors):The resubmission has corrected some of the issues I raised previously. However, I have some remaining concerns, primarily about the interpretation of the proposed causal effects and the simulation results.

Thank you for your positive assessment of our manuscript.

1. My largest issue is still in the causal interpretation of genetically determined PCs are sparse components. To me, I don't think there is a way to turn the components into well-defined exposures that admit a counterfactual or interventional definition of a causal effect. This approach seems closer to "causality adjacent" methods like TWAS. There was a paper a few years ago using sparse CCA in combination with TWAS (https://journals.plos.org/plosgenetics/article?id=10.1371/journal.pgen.1008973) and this seems similar to that to me. I think one option for this paper would be to add a deeper discussion of what it means to test for a causal effect of a component that may or may not represent a latent biological variable. On one end of interpretation you have "association with genetically predicted exposure component" and then I think you could add as a bonus that if you have successfully identified a biological component, this could be interpretable as a causal effect.

Thank you for the article suggestion, we have read the particular application of sparse methods in TWAS and cite it in the discussion. This also helps contextualising the benefits of sparse methods in general, namely improving power.

We have supplemented our previous discussion of this difficulty in interpreting the components as actionable exposures with counterfactuals in the draft, in the Discussion/ Limitations section, bringing it forward in the new updated paragraph. We acknowledge that a strict causal interpretation is not readily available with our suggested approach; at the same time, we believe it’s highly unlikely that a pragmatic causal effect will act in a way such that only one particular exposure of the many, highly correlated ones will be the sole driver of an effective medical intervention.

2. Using the components introduces an issue of horizontal pleiotropy if a single variant is allowed to affect multiple components. This suggests that the component-based MVMR results would be more appropriate than component-based UVMR results, though for PCA this is not an issue due to orthogonality and approximate equivalence of UVMR and MVMR.

Thank you for your comment. We agree that MVMR results are better suited for this purpose and this is why we present MVMR and not UVMR in the positive control example with real data. We highlight this in the applied Results section and mention the pleiotropy concern in the limitations.

Some questions about the simulations:3. In Figure 4, it seems like the MVMR plot should have 6 lines and not two. How were the 6 traits combined into two lines here? It seems unfair to average the rejection rate for the X_1-3 into the black line since, if MVMR performed perfectly, it would never reject X_3 and always reject X_1 and X_2.

Thank you for your comment. We have substantially revised the simulation study results, splitting it in three sections to improve clarity (Illustrative example, high-dimensional example, determinant of performance). The changes are highlighted in blue. We have also corrected the data generating mechanism in Figure 4a, to visualise the exact mechanism that was used in the simulations in Figure 4b (six exposures forming two blocks, one block / three exposures are causal for Y).

In the first scenario (Figure 4), we have compared the rejection rates of MVMR with the two informative PCs and MVMR with the six individual exposures. Therefore, the first two panels in Figure 4b include two lines (with corresponding Monte Carlo SEs), one for each PC, and the rightmost panel includes six lines, one for each exposure. We have updated the colour coding and the DAG to better present this.

In the proposed workflow, we have used only the informative components in order to efficiently reduce the dimensions. We achieved this with permutations in the real data and with the Karlis-Saporta-Spinakis criterion, a simpler method for a different eigenvalue cutoff. In the simulations, we have used the two first PCs, each of which reflected three exposures.

4. Why is there an interval around only the black line in the MVMR plot and not the other plots?

We have now modified the presentation of the Figure. What we are presenting in this plot is the rejection rates for the two components (three exposures each) of PCA and SCA. In MVMR, there are six exposures and six rejection rates. The colour-coding is green for the true causal exposures (and components) and red for those that do not affect the outcome. To make this clearer, we now show different shades of red and green for the individual exposures and red and green for the transformed components. For PCA, the causal and the non-causal *groups* are presented as two separate lines. It was observed that PCA and SCA performed well in mapping the correlated exposures to two components, hence the equivalence visualised by the common colour coding (see also comments 3, 6). Apologies for the confusing presentation, the lines that were interpreted as intervals are in reality lines that show the rejection rates of MVMR for each exposure. We have added a table in the figure to clarify this, and updated the figure legend.

Therefore, an ideal performance would be a nominal type I error rate for the component that reflects the negative exposures (colour-coded red) and a high value of power approaching 1 (colour coded black). To improve presentation, we have also added Monte Carlo SEs following Tim Morris et al. (Table 6 in DOI: 10.1002/sim.8086). In this updated simulation, there seems to be an appropriate performance for SCA, with T1E converging at 5%. The problematic performance of MVMR is highlighted in the lower than nominal Type 1 Error throughout and, importantly, in the low power in the low instrument strength simulation (CFS ~ 3, first result/ leftmost part of the three panels).

5. The sentence in the caption presumably describing the color scheme is cutoff. "The results are split according to group n black,…."

Thank you for your comment, we have now rephrased the Figure 4 caption.

6. Is the appropriate number of components known a-priori in the simulations?

This is correct. The two ways of determining the number of components (KSS criterion, permutation) performed well in the simulated data and we chose to present results with an a priori known number of components. We have added this in the simulation results/ Simple Illustrative Example section for transparency.

7. The description of Figure 4 suggests that PCA and SCA are out-performing MVMR. However, PCA has exactly the same rejection rate for the causal and non-causal block of traits which seems like bad performance to me. SCA has a Type 1 error rate between 20 and 30% at a nominal level of 0.05 which also seems quite bad to me. I would not view improving power at the cost of uncontrolled type 1 error as a good outcome.

Thank you for your suggestions, we have given the simulation results a more careful consideration. We agree that PCA performs suboptimally, albeit in a different manner than MVMR. MVMR seems to lack in power and PCA provides many false positive results. What seems to perform better is SCA. We also agree that a T1E of 20% is concerning. However, observing that the T1E curve didn’t seem to have fully converged for the instrument strength that we were previously reporting, we repeated the same simulation study with one modification. We further increase the sample sizes to achieve higher instrument strength. In the updated Figure 4, SCA indeed outperforms PCA and MVMR, with a well-controlled T1E for larger samples. It may be unlikely that such a high instrument strength will be observed, especially in the context of correlated exposures, however we believe this convergence to 5% is quite reassuring.

Regarding your observations on the performance of PCA, we agree and we are more critical of PCA in the Results section as we agree that false positives are a major issue. It seems that MVMR and PCA in the formulations we investigated perform poorly at the two extremes of poor performance (highly imprecise, misleadingly precise); meanwhile, sensible transformations of the exposures are achieved and we have added in the discussion the possibility of some form of T1E control to salvage PCA performance to guide further research. To clarify this, we added a segment in the discussion (Paragraph 2 ‘When using PCA without any sparsity constraints, our simulation studies revealed numerous false positive results, at the opposite end of the nature of poor performance seen in MVMR; estimates were often misleadingly precise (false negative). Although appropriate transformations of the exposures were achieved, we highly recommend exploring additional forms of T1E control to improve the performance of PCA. Nonetheless, sparse methods exhibited superior performance compared to both PCA and MVMR.6’)

8. The description of the Figure 5 at the top of page 9 does not align with the image in Figure 5. "Both standard and Bonferroni-corrected MVMR performed poorly in terms of AUC (AUC 0.660 and 0.885 respectively), due to poor sensitivity. While there was a modest improvement with PCA (AUC 0.755), it appears that PCA and RSPCA do not accurately identify negative results (median specificity 0 and 0.28 respectively)." Reading these numbers, it sounds like MVMR is the worst followed by PCA, followed by MVMR_B. However, in the figure, PCA and RSPCA both have lower AUC than MVMR and MVMR_B. These numbers are also inconsistent with Table 4.

Thank you for your comments, the purpose of direct AUC comparisons is to visualise in one figure the performances of the methods. We agree that, seen in isolation, it may be misleading to miss the differences with respect to sensitivity and specificity across methods: MVMR consistently provides imprecise results and fails to identify positive associations, whereas PCA provides very precise results that yield many false positives. We were acknowledging this previously and have further highlighted in the discussion. The comparison of PCA and MVMR solely on AUC is therefore not a reliable way to measure their efficacy. Thank you for spotting the inconsistency in Table 4, the columns are now correctly labelled.

At the same time we present the results, we’re discussing what is not directly identified. Results are driven by very different behaviour on different ends of AUC curve. We acknowledge that the results are inherently dependent on the data generating model and the specific simulation settings used in our study. If the data were generated differently, the results would differ as well.

9. I think I may be a little confused about how the ROC curves are constructed.9a. In my experience, an ROC curve is usually constructed by varying a threshold (e.g. α). However, it seems here like α was fixed and we are averaging over the single point obtained for each simulation setting according to the Reitsma et al. method. This method makes sense when the data are real observations, however, in the case of simulations, why not use the traditional method of varying to threshold and then averaging the full curves across simulations?

Thank you for bringing up this question. We have substantially updated the simulation section and we hope the workflow is now clearer. We first present the data generating mechanism, and we provide more details on the ROC curve construction in the paragraph Simulation Studies/ Complex High-Dimensional Example.

We fix the α threshold to present the results across many individual simulations in a more concise manner. Varying the threshold and averaging the full curves across simulations would have been a valid approach for single simulation studies as well, but any summary across simulation studies would have made the presentation of results more complex and difficult to interpret.

Therefore, we use the Reitsma et al. method, a well-established approach for summarizing ROC curves in simulation studies, as it provides a comprehensive and concise summary of the results. This method jointly meta-analyses sensitivity and specificity with a bivariate mode. We further explain our rationale in the Simulation Studies/ Complex High-Dimensional Example paragraph.

9b. I think this method of displaying results also obscures the issue of calibration. If there is a nominal significance threshold, we would like to see that type 1 error is controlled below that threshold.

Thank you for your comment. We have shown that SCA achieves appropriate T1E for larger samples in the updated analysis (Simulation Results/Simple illustrative example/Figure 4).

9c. Visually, Figure 15 looks inconsistent with Figure 5 to me. There are no blue points in the upper right of Figure 15 -- does it make sense to extrapolate this part of the curve? Figure 5 suggests that the red methods at least have the potential to do very well in differentiating true and false positives but this doesn't seem born out by Figure 15 where most of the red points cluster in the upper right.

Thank you for your observation. The curve is directly derived from the bivariate meta-analysis method and the unobserved values are extrapolated from the form of the observed ones. The true performance, as observed directly from individual simulation studies, is reported in Figure 15, where many of the results for the two superior sparse PCA methods cluster in the top right corner. Additionally, there is a clear benefit in the red methods, as evidenced by a substantial scoring in the top of the y-axis (in the sensitivity = 1 area of the plot). This benefit is not observed for PCA or MVMR. When the results are meta-analysed, these latter simulation studies in the middle region are extrapolated and indicative of good performance. This is why we choose to meta-analyse, as all these factors are taken into account, but at the same time present the individual results for visual comparison.

We have to disagree with your interpretation regarding how these two results are in disagreement, as we believe Figure 15 is a transparent and complementary way of presenting the total performance in Figure 5. In Figure 15, even though there is some apparent clustering in the top right corner, there is still much more dispersion compared to PCA and especially in the middle regions (say sensitivity and specificity ~ 0.2 -0.8). These individual results give rise to the curves in the meta-analysed plot.

10. On the selection of instruments -- is it correct that you are assuming that all variants affecting any of the component metabolites also affect LDL/HDL or Triglycerdies as measured in the GLGC study? Does this assumption limit your ability to identify components that separate LDL or HDL sub-components? For example, variants that affect only a small number of components will be less likely to be selected using the GLGC reference.

This is true, ideally we would have an external study that investigates sub-components as separate phenotypes rather than serum measurements of total fractions. We are acknowledging this as another limitation in the discussion.